

# Operationalizing fine-scale soil property mapping with spectroscopy and spatial machine learning

Thorsten Behrens[1,2], Karsten Schmidt[1], Felix Stumpf[1], Simon Tutsch[1], Marie Hertzog[1], Urs Grob[1], Armin Keller[1], and Raphael A. Viscarra Rossel[3]

[1]Swiss Compentence Center for Soil (KOBO), School of Agricultural, Forest and Food Sciences (HAFL), Bern University of Applied Sciences, Länggasse 85, 3052 Zollikofen, Switzerland
[2]Soil and Spatial Data Science, Soilution GbR, Heiligegeiststrasse 13, 06484 Quedlinburg, Germany
[3]Soil & Landscape Science, School of Molecular & Life Sciences, Faculty of Science & Engineering, Curtin University, GPO Box U1987, Perth WA 6845, Australia.

**Correspondence:** Thorsten Behrens (thorsten.behrens@bfh.ch)

**Abstract.** One challenge in soil mapping is the transfer of new techniques and methods into operational practice, integrating them with traditional field surveys, reducing costs, and increasing the quality of the soil maps. The latter is paramount, as they form the basis for many thematic maps. As part of a novel approach to soil mapping, we integrate various technologies and pedometric methodologies to create soil property maps for soil surveyors, which they can utilize as a reference before

beginning their pedological fieldwork. This gives the surveyors considerably more detailed and accurate prior information, reducing the subjectivity inherent in soil mapping. Our approach comprises a novel soil sampling design that effectively captures spatial and feature spaces, mid-infrared spectroscopy, and spatial machine learning based on a comprehensive set of covariates generated through various feature engineering approaches. We employ multi-scale terrain attributes, temporal multi-scale remote sensing, and Euclidean distance fields to account for environmental correlation, spatial non-stationarity,

and spatial autocorrelation in machine learning. Methods to reduce the uncertainties inherent to the spectral and spatial data were integrated. The new sampling design is based on a geographical stratification and focuses on the local soil variability. The method identifies spatially local minima and maxima of the feature space, which is fundamental to soil surveys at the specified scale. The k-means and Kennard-Stone algorithms were applied in a sequential manner within each cell of a hexagonal grid overlaying the study area. This approach permits a systematic sub-sampling from each cell to analyze predictive accuracy for

varying sampling densities. We tested one to three samples per hectare. Our findings indicate that a sample size of two samples per hectare was sufficient for accurately mapping soil properties across 300 hectares. This markedly reduces the financial burden associated with subsequent projects, given the significant reduction in the time and resources required for surveying. The spectroscopic and spatial models were unbiased and yielded average $R^2$ values of 0.91 and 0.68–0.86, depending on mapping with or without pedotransfer models. Our study highlights the value of integrating robust pedometric technologies in

soil surveys.





## 1 Introduction

The base maps and information typically available to surveyors for planning and executing a soil survey comprise data on soil-forming factors, including terrain, land use, climate, and, when available, parent material. In most cases, soil maps from previous surveys are unavailable. As a consequence of the limited specificity of the initial data set, during a field campaign, the surveyor's mental model undergoes a process of "Gestalt shift," whereby the landscape and the soil formation processes within it come to be understood (Hudson, 1992). This phenomenon is associated with the surveyor's tacit knowledge and personal experience (Polanyi, 1966; Hudson, 1992). In this context, we propose a framework to generate soil property maps for soil surveyors to use in their pedological fieldwork. The use of detailed soil property maps, including data on sand, silt, clay, soil organic carbon (SOC), carbonate content, and pH value, as a basis for the construction of a mental model of the soil landscape before fieldwork can help to minimise the extent of any 'Gestalt shifts' and the potential for subjectivity in the results. This allows the surveyors to focus on the core elements of their expertise, which sensors cannot adequately capture, such as the impact of soil moisture and pedogenesis. For such an approach to be cost-effectively implemented over large areas, the density of sample locations must be low while ensuring sufficient coverage to capture soil variability.

Generating modern, digital soil property maps comprises a robust soil sampling and sensing design, rapid sample preparation, sensing, measurements with conventional methods, soil spectroscopy, feature engineering, and spatial modeling with machine learning. This provides the higher density of analytical data required by end users to generate thematic maps and significantly improves the quality of the final soil maps and thematic products. Therefore, the primary objectives of the work presented here are to describe the framework for soil property mapping using soil sensing and machine learning and to introduce a new sampling design that accounts for multi-scale soil variation.

This work is embedded within a broader conceptual framework for integrated fine-scale digital and analog soil mapping developed by the Swiss Competence Center for Soil. Based on a series of case studies, the objective is to standardise and further develop soil mapping methods, to transfer new techniques and methods into operational practice, and to integrate them with traditional field surveys. The overarching objective is to reduce the costs of high-resolution soil mapping and to enhance the quality and applicability of the soil maps produced. The work conducted by the Swiss Competence Center for Soil in integrated operationalization projects is carried out at scales between 1:5,000 and 1:10,000.

The local variability of soils in Switzerland is high, often driven by differences in soil moisture, which is relevant to soil quality. Therefore, the spatial density of pedological field recordings must be high and evenly distributed (Siegrist and Marugg, 2023; AfU Solothurn, 2024). The typical density of pedological recordings in a soil mapping campaign in agricultural areas relevant to food production in Switzerland is four locations per hectare (100×100 m), with one location documented by the surveyor. Despite the use of efficient drilling vehicles and sample preparation techniques, the costs to gather samples for laboratory analysis at this sampling density remain high. Consequently, one objective of the operationalization projects is to analyze the effects of spatial sampling density on predictive accuracy.

Depending on the survey areas, which are often based on administrative boundaries, it is quite likely that the feature space of covariates relevant for a stratified soil sampling design, such as terrain attributes, does not provide even geographical stratifi-



cation or spatial coverage (Brus, 2022). Sampling design algorithms often prioritise regions within the feature space with high variation. As a result, regions that exhibit variability receive more samples disproportionate to their actual size. Such regions may be relatively small, for example, 5% of the total area, and therefore of little relevance in the context of the soil map. As a result, gradients and local variability in larger, more homogeneous areas may be underrepresented because a sparser sample set does not adequately cover them. In addition, sampling designs such as k-means and Kennard-Stone tend to identify new

transition zones when the extremes of the covariates are already covered. This may be significant in certain cases, but in the case of soil mapping, most relevant local extremes are omitted in favor of transition zones between the extremes. Pedologically, this is not relevant in most cases where, for example, local differences in soil moisture are most relevant to soil classification, and local surface highs and lows should be surveyed to account for such differences. To address both of these issues, i.e. spatial coverage and local variability, a geographical stratification sampling approach has been developed that focuses on local

variability in feature space. The k-means and Kennard-Stone algorithms were applied sequentially within each cell of a hexagonal grid covering the study area. This approach allows for a systematic sub-sampling from each cell, enabling the analysis of predictive accuracy for varying sampling densities. A further advantage of the proposed design is that it can be systematically sub-sampled during modeling to analyze the effect of sample density, which is important to further reduce costs for similar studies in the future.

## 2  Methods

This section describes the study area, the feature engineering approaches used, the multi-step sampling design for mapping and calibration, the laboratory and field work, and the modeling approaches applied. The general workflow of the framework presented here is shown in Figure 1. Starting with the spatial feature engineering required for the sampling design and spatial modeling, a major part consists of analyzing covariates for the sampling design. After both sampling designs for soil

spectroscopy and reference analysis, spectral and spatial modeling with different sampling densities follows.

### 2.1  Study area and number of sampling locations

The study area is situated in the Swiss canton of Fribourg and encompasses an area of approximately 300 hectares in Prez-vers-Noréaz (municipality of Prez) (Figure 2). Exclusion areas were defined before deriving the soil sampling design, including roads, drains, gas pipes and residential areas. This resulted in an area of approximately 220 hectares that could be sampled.

### 2.2  Feature engineering and covariate selection

#### 2.2.1  Covariates for spatial modeling

Several feature engineering techniques were employed to derive the necessary covariates for the subsequent spatial modeling. The terrain attributes were derived from the SwissALTI3D digital elevation model (DEM), provided by the Federal Office of Topography 'swisstopo', with a resolution of 0.5 m. The road network was removed from the DEM, and any resulting gaps



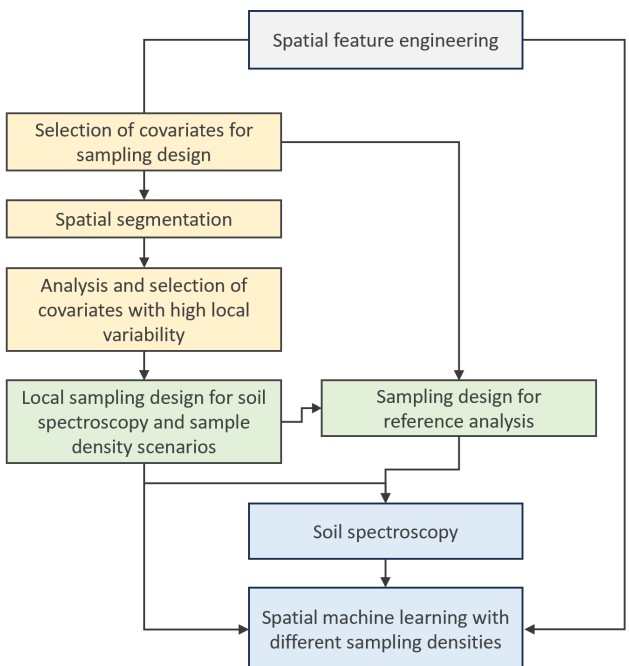

**Figure 1.** Overview of the data and processing steps.

were interpolated for the terrain analysis. The DEM was downscaled to a resolution of 2 m for modeling. For all regional terrain attributes, such as flow accumulation, depressions were filled (Wang and Liu, 2006) and small dams were breached (Lindsay, 2015) using SAGA GIS (Conrad et al., 2015).

Gaussian mixed scaling (Behrens et al., 2018a), was used to derive elevation, local terrain attributes (slope, eastness, northness, mean curvature, planform curvature, profile curvature (Zevenbergen and Thorne, 1987)), and the regional terrain attributes flow accumulation (Qin et al., 2011) as well as the SAGA topographic wetness index(Böhner and Selige, 2006)) on nine scales. In Gaussian mixed scaling, the resolution is reduced by half with each consecutive octave or scale. Furthermore, three additional intermediate scale levels have been calculated between the octaves (Behrens et al., 2018a). This yielded 40 scales per terrain attribute and 360 terrain derivatives with internal resolutions between 2 m and 1024 m. Furthermore, local kernel-based flow accumulation and terrain position (Behrens et al., 2024) were derived on five different neighborhood sizes, from 25m to 500m, with five different local noise and sink removal settings. This resulted in 50 additional terrain covariates. Euclidean distance transforms were derived for the geographic coordinates, as well as a grid of 35 (5 by 7) points spanning the extent of the study area, to facilitate spatial interpolation with machine learning (Behrens et al., 2018b; Behrens and Viscarra Rossel, 2020).

The vegetation characteristics are subject to change over time and across a range of temporal scales. This depends on weather and climate fluctuations, land management practices, soil properties and the characteristics of the parent material (Heuvelink et al., 2021; Venter et al., 2021; Wang et al., 2018). The temporal variability of vegetation can be described using land use



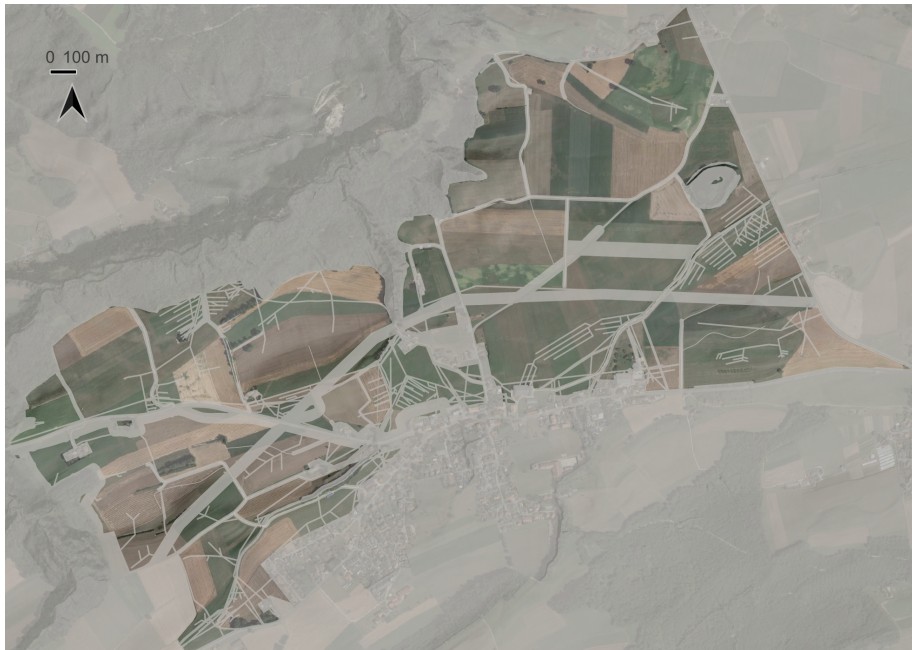

**Figure 2.** Study area showing also inaccessible regions for sampling, including roads, drains, gas pipes and residential areas. A satellite image (©2022 Google) is superimposed on a hillshade model in the background.

covariates based on a seasonal vegetation index, which is processed for each year and across multiple time scales. This study employs a comparable methodology based on annual aggregations utilizing the median and standard deviation of an NDVI raster time series derived from Landsat imagery(Stumpf et al., 2024). The temporal scales were calculated for 15 time spans

by aggregating the Landsat data over five years in each case, spanning 1989 to 2021. Furthermore, a monthly NDVI raster time series based on Sentinel 2 imagery was used to describe the temporal variability of vegetation. The NDVI of each month was averaged across the years 2018 to 2022, and the NDVI standard deviation of each year was averaged across the same period. In total, 153 land use-related covariates were derived.

Data representing the spectral signature of vegetation-free surfaces, referred to as "bare soil," was extracted from the Landsat

time series. Therefore, all pixels from all individual images between 1985 and 2022, which were identified as bare surfaces, were used to create a median composite. This composite describes the soil reflectance in the visible and infrared range in six bands for each pixel. Bare pixels were identified through the application of NDVI values < 0.25 and NBR2 values < 0.0075, with the exclusion of green vegetation, straw, crop residues and high soil moisture content(Safanelli et al., 2020; Sorenson et al., 2021; Castaldi et al., 2019; Urbina-Salazar et al., 2023; Stumpf et al., 2024). In areas with permanent vegetation (e.g.,

forest areas, permanent grassland), bare soil reflectance was extrapolated using machine learning (Stumpf et al., 2024).





The land use variability and bare soil data were upscaled from the Landsat images' original spatial resolution of 30 m to the terrain attribute resolution of 2 m. This was achieved through spatial modeling with machine learning. The land use variability data from Sentinel were resampled from 10 m to 2 m using bicubic spline interpolation.

### 2.2.2 Covariates for sampling design

The sampling design was based on a combination of carefully selected uncorrelated covariates to describe the distribution of pedological features within the study area. Accordingly, we selected a local and a regional version of the kernel-based flow accumulation (50 and 500 m, (Behrens et al., 2024)) to provide information on the local and regional soil moisture patterns, the SWIR2 Baresoil dataset of the Landsat data, which is related to mineralogy and has been demonstrated to be relevant for pedological models (Stumpf et al., 2024), and the Sentinel 2 and Landsat variability data described above, which address the

influence of land management. Furthermore, we derived the topographic roughness (Riley et al., 1999) as a terrain attribute, which focuses on local surface variability and indicates local minima and maxima. The corresponding datasets are presented in Figure 3. To ensure that the covariates were transformed into a normal distribution, as required for the parametric statistical sampling design approaches used in this study, the non-parametric transformation method of "ordered quantile normalization" was employed (Peterson and Cavanaugh, 2019).

### 130 2.3 Sampling design

In this study, three sampling locations were selected per hectare. The aim is to obtain a representative sample of the local variability of the soil evenly across the whole geographical space. Maintaining a relatively uniform sampling spacing is essential to ensure comprehensive coverage and to prevent the potential for small-scale geological or anthropogenic influences to be inadequately recorded.

We thus propose a two-stage process for selecting sampling locations. The first stage comprises segmenting the geographical space and selecting relevant, explicitly locally variable covariates. The second stage is the locally adaptive sample design, conducted separately within each spatial segment based on the selected covariates. The following sections describe the two stages in further detail.

### 2.3.1 Geographical stratification

The sampling design is based on systematically segmenting the geographical space into hexagons. One of the advantages of a hexagonal grid is that all neighbouring centres are situated at an equal distance from each other, which is not the case with a square grid. This results in a more uniform distribution in geographical space. In contrast to stratification using cluster analysis, all hexagons are of an identical size. In this study, the size of the hexagons was set to one hectare 4. This allows spatially uniform sub-sampling and correspondingly straightforward evaluation of the required sampling density. This geographical

stratification facilitates the detection of local minima and maxima in the sampling design, which is conducted separately





**Figure 3.** Transformed input data for the sampling design.

within each hexagon. A buffer zone of four metres was applied to prevent edge artefacts, such as directly adjacent sampling sites at the hexagon boundaries.



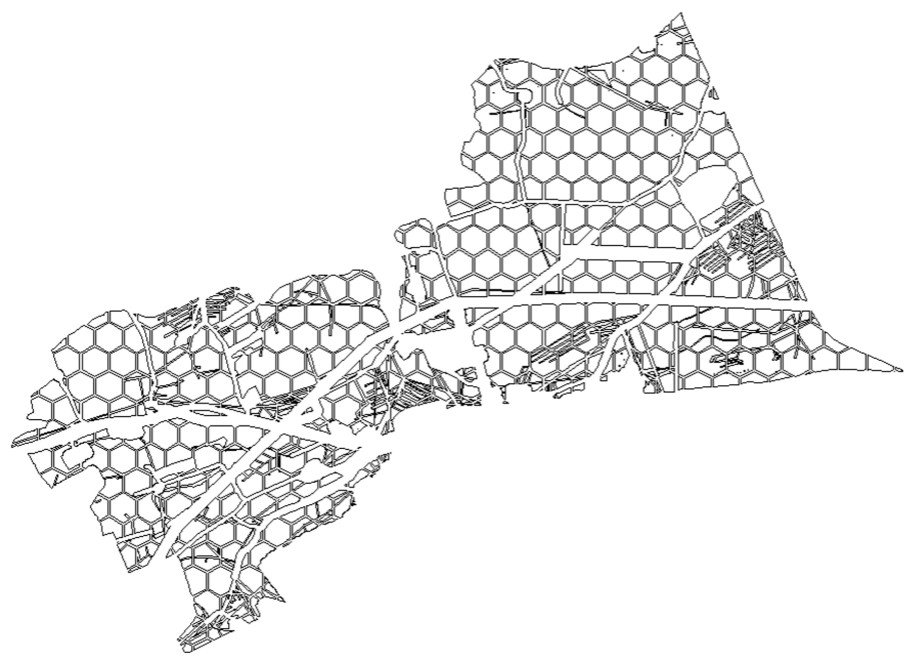

**Figure 4.** Hexagonal grid for the spatial distribution of sampling points.

Only covariates that exhibited local variation were used to select the most appropriate sampling points within each hexagon. This includes, for example, local variations in elevation or depression, which may indicate a different water balance, material translocation, or local variations in soil color or land use. In contrast, covariates that are less variable across the landscape generally have less impact on the local soil variability (Behrens et al., 2019; Behrens and Viscarra Rossel, 2020). We therefore hypothesise that their distribution is already covered by the close-meshed, locally adaptable design and thus need not be considered at this stage. We will test this assumption by comparing the frequency distributions of the covariates and their values in the sample set.

The analysis of the local variability of the covariates and the evaluation of their relevance in the sampling design were conducted as follows. First, the standard deviation of each covariate within each hexagon was calculated as a measure of local variability. Secondly, the standard deviations of all hexagons were sorted. If the standard deviation of the covariates in the hexagons is low, it indicates the absence of relevant local variability, and those covariates are not selected. Consequently, only those covariates showing a standard deviation of 0.5 in the transformed value space in at least 30% of all hexagons in the study area were selected.

**Local sampling design**

We propose combining two methods to select locations that cover local minima and maxima in feature space while also ensuring the exclusion of extremes that may result from irrelevant noise in the covariates. We will then derive three locations





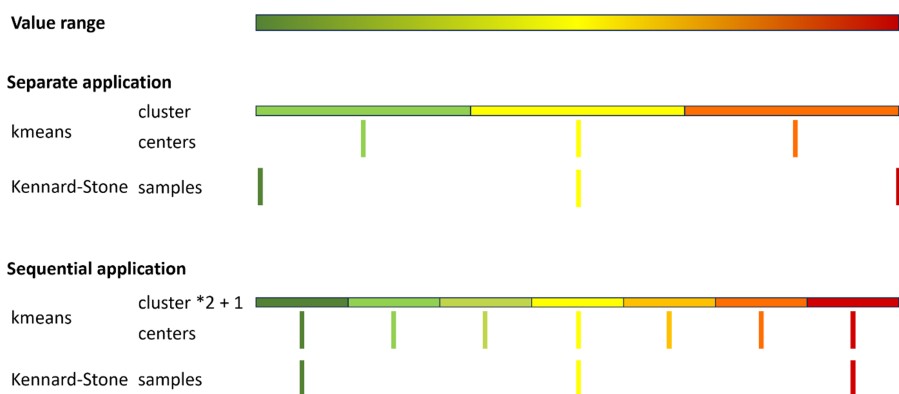

**Figure 5.** Conventional separate application and proposed sequential application of k-means and Kennard-Stone sampling for identifying three samples from a feature space (green to red).

separately for each hexagon. This approach permits a systematic sub-sampling of three, two, or one sample from each cell to

analyze predictive accuracy for different sampling densities.

In the initial phase, k-means sampling (Steinhaus, 1956) is employed, whereby the number of samples, designated as m (set at three in this study) is increased by a factor n and the value p (calculated as m*n + p) to generate a larger and odd number pool of clusters that delineate distinctive combinations of the input data. Expanding the cluster set through oversampling produces a more detailed representation of the feature space and a greater divergence between the selected local minima and maxima,

thus enhancing the possibility of individual clusters encompassing relevant extreme zones of local variability. The proposed approach identifies clusters relevant to the local context and functions as a filter by grouping multiple pixels, reducing the likelihood of selecting non-relevant noisy pixels as samples from the covariate data.

The final three sampling locations are then selected from this extended set of potential locations using Kennard-Stone sampling (Kennard and Stone, 1969). This set thus comprises two locations that are the most distant from one another in the

feature space and a location that represents a more average position in the feature space. Figure 5 illustrates the selection of three sites with k-means and Kennard-Stone sampling in a typical separate application over a hypothetical feature space, as well as the sequential approach presented here, whereby a Kennard-Stone sample is drawn based on a sample set drawn by k-means.

In the case of incomplete hexagons resulting from the presence of roads, settlements, or the boundaries of the study area, only

two or one sites were selected, with the number of samples depending on the remaining hexagon area. Incomplete hexagons were assigned a maximum of two sampling locations if they had an area between 20 and 60 %. This did not apply to hexagons where soils were drained, and the drainage network was buffered and clipped out. A high degree of pedological variability was anticipated in these hexagons, and three sampling sites were selected. Sampling was not conducted in highly fragmented or marginal areas with sizes of less than 20 %.



The local sampling design was adapted for instances where only one or two samples could have been placed in the remaining area. In the case of the two remaining samples, Kennard-Stone sampling was employed to pick the two extremes. In cases where only one sample was required, the average location was selected based on k-means sampling with only one cluster. The final sample set comprised 812 locations.

**Alternative sample area**

An alternative area was calculated for each of the 812 sampling locations, within which samples could be taken if the original locations were inaccessible. The new location should be as proximate as possible to the designated point. The Euclidean distance in the feature space was the basis for creating the alternative areas.

### 2.3.2   Sample selection for reference analysis

To develop spectroscopic models of soil properties, a subset of the samples must be analyzed using conventional laboratory
techniques (see below). This subset of samples for reference analysis should represent the variability of the entire area. Therefore, to select this subset of samples, we included covariates that exhibit variability at larger spatial scales and those exhibiting local variability.

To determine the most appropriate sampling design approach for deriving the reference locations, we drew independent samples using both the k-means and Kennard-Stone sampling methods. To evaluate the representativeness of the two sampling
designs for covering the feature space of the underlying population, scatter plots, their convex hulls, and density functions were constructed for different combinations of two input covariate datasets. An optimal sampling design ensures the most precise representation of the population in terms of the density function and the coverage of the feature space. The sample sets derived using k-means and Kennard-Stone both comprise 45 locations.

### 2.4   Field and laboratory work

At each location, soil samples were taken from three depths: 0–20 cm, 20–40 cm and 40–70 cm. We used an automated sampler to facilitate efficient field sampling and operationalisation. Upon removing the auger from the soil, the soil material is automatically transferred into distinct containers, each corresponding to a predefined depth interval. Sampling from three depths is completed in approximately 30 seconds, allowing for sampling of approximately 75 sites per day, depending on the accessibility and spacing of the sampling locations. Ultimately, 810 samples were obtained from the 0–20 cm interval, 801
from the 20–40 cm interval, and 728 from the 40–60 cm interval.

The samples were packaged and provided with a QR code before being dispatched to the laboratory for preparation and measurement. They were then dried at 105°C. Grinding was conducted in a grinding rack capable of processing up to 100 samples in parallel. A total of 255 samples were selected for analysis, which included measuring soil pH in a 0.01M $CaCl_2$ solution, determining clay, silt, and sand contents by sedimentation, quantifying organic carbon content using the thermal
gradient method, and measuring carbonate content using the volumetric method.





The MIR spectra of the 2339 samples were recorded with the Invenio FT-IR spectrometer (Bruker Optics, Germany) at 2 cm$^{-1}$ spectral resolution and 64 scans per sample. Four replicates were measured for each soil sample. Replicates that showed inconsistencies were identified based on the Euclidean distance between the spectra and subsequently removed. The remaining replicates were averaged to generate a single spectrum per sample.

## 2.5 modeling

### 2.5.1 Spectroscopic modeling

The spectroscopic modeling was based on stacking base learners via a generalised linear model (GLM) in the statistical software R (R Core Team, 2022). The following base learners were used: Cubist (Quinlan, 1992; Kuhn and Max, 2008), extreme gradient boosting (Chen et al., 2024), bagged multivariate adaptive regression splines (Friedman, 1991; Kuhn and Max, 2008; Milborrow, 2024) and radial basis function support vector machines (Karatzoglou et al., 2004). The hyperparameters of the models were optimised using the R packages caret (Kuhn and Max, 2008) and caretEnsemble (Deane-Mayer and Knowles, 2023). The evaluation of all models was conducted using 5 times 10-fold cross-validation. The Gaussian pyramid approach was applied to derive the first derivative of the spectra at six different resolutions (Behrens et al., 2022). Models were built for all resolutions to determine the optimum resolution and independently model each soil property.

### 2.5.2 Spatial modeling

As with the spectroscopic modeling, a spatial model was created using stacking via a generalised linear model (GLM). The same cross-validation procedure was applied, and the same base learners plus random forests were used (Breiman, 2001; Grimm et al., 2008; Liaw and Wiener, 2002). Over 600 spatial predictors were included as covariates.

Following an initial modeling run for all soil properties and for each depth, a second run was conducted, incorporating the previous models in terms of a pedotransfer function. The corresponding data for each soil property from the previous run was excluded to avoid circular reasoning. The rationale behind this approach is to leverage knowledge derived from analyzing a given soil property to facilitate utilizing that knowledge for other soil properties for which the relationship may not yet be apparent but may prove relevant. To address potential uncertainties in the coordinates of the sample locations, which could arise from GPS positional inaccuracies or data inconsistencies, a random corruption analysis (Grimm and Behrens, 2010) was employed, which returns updated coordinates for each sample location. The analysis was conducted at a narrow spatial radius of two pixels or 4m.

### 2.5.3 Scenarios of varying sampling densities

Three sampling densities were compared with the spatial modeling. The first was all three samples per hexagon, expected to capture the local spatial variability in the greatest detail. The second was only the Kennard-Stone algorithm's two extreme samples selected from the k-means sample set. The third was only the Kennard-Stone sample furthest away from the other





two locations in the feature space. The spatial modeling was run separately for each set, and the predictive accuracies were compared to evaluate the impact of the sampling density and inform the design of future projects.

## 3   Results and discussion

### 3.1   Spatial segmentation using hexagons

Figure 6 illustrates the spatial analysis of local variability for each hexagon and input data set based on the standard deviation. The ordered standard deviations of all hexagons for each input data set are presented in Figure 7. Based on the defined threshold value for local variability, the final parameters selected were the local flow accumulation, the topographic roughness, and the land use variability based on the sentinel data, as these exhibited a standard deviation of the normalised data of at least 0.5 for 30% of all hexagons in the study area.

### 3.2   Sampling design for modeling

Figure 8 shows the final sampling design, the hexagon boundaries and the alternative sampling areas. The sampling densities per remaining hexagon area are presented in Figure 9

A comparison of the frequency distributions of the input data set (grids) with that of the selected sampling locations reveals a high degree of correspondence (Figure 10). Due to the emphasis on local, small-scale variability, the marginal regions of

the sample set distributions show a slight increase in density. This results in slightly more uniform coverage, which can help to prevent the suppression of these underrepresented value ranges in the modeling process. From a thematic perspective, this falls within the domain of imbalanced learning (Branco et al., 2017; Taghizadeh-Mehrjardi et al., 2020; Camacho et al., 2022). Imbalanced learning methods aim to achieve a more balanced distribution of data points in a given dataset before modeling. This is achieved by reducing the number of sampling points within heavily populated value ranges and generating additional

synthetic sampling points for underrepresented value ranges. In contrast, the proposed sampling method utilises actual data points from the underrepresented value ranges.

### 3.3   Sampling design for reference analysis

The samples utilized for the reference measurements are derived from the sample set designated for spectroscopic and spatial modeling. The objective is to ensure that the reference samples encompass the full feature space pertinent to the area under

consideration.

As illustrated in Figure 11, both approaches tested have inherent limitations. The Kennard-Stone sample set covers the feature spaces almost completely; however, the density functions generally exhibit a much more uniform and sparse distribution. In contrast, the k-means sample set shows density functions that align more closely with those of the populations, yet they do not fully span the feature spaces, as evidenced by the smaller convex hulls.



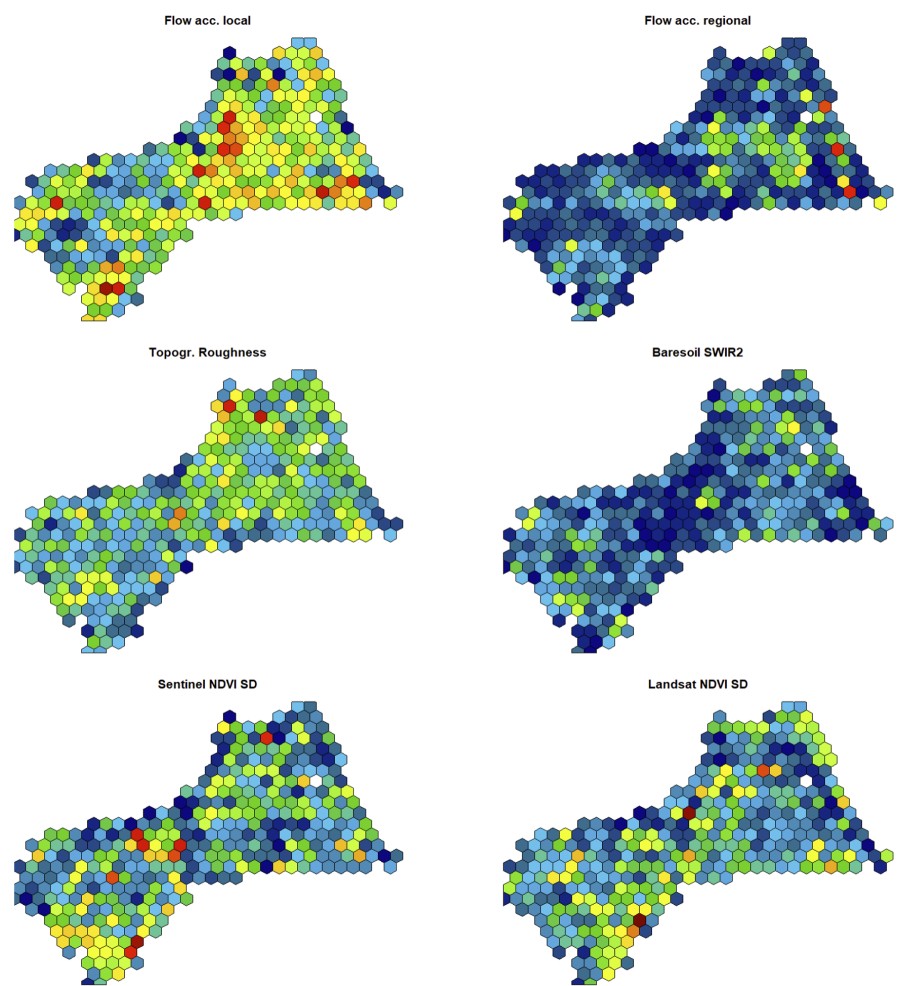

**Figure 6.** Variability (standard deviation) of the potential input data per hexagon (blue: low; red: high).

While Kennard-Stone sampling ensures uniform and full coverage of the feature space, k-means identifies distinctive combinations within the input data set. These properties are integral to the representativeness required to cover the covariate feature space optimally. Figure 12 compares the frequency distributions of the entire input data set with the two sample sets and their combination. This illustrates that the local sampling methodology effectively encompasses the feature spaces of data sets with larger spatial scales (Baresoil, Flow acc. regional), which were not incorporated into the local sampling design within each hexagon. This supports the hypothesis previously proposed, namely that the feature spaces of covariates with larger scales, which are irrelevant for local variations, are adequately covered by the derived fine sampling density. Moreover, Figure 12 illustrates that the integration of both sampling design methodologies most accurately represents the frequency distributions of the population (the sample set for spectral and spatial modeling). Consequently, the combination of sampling sites selected





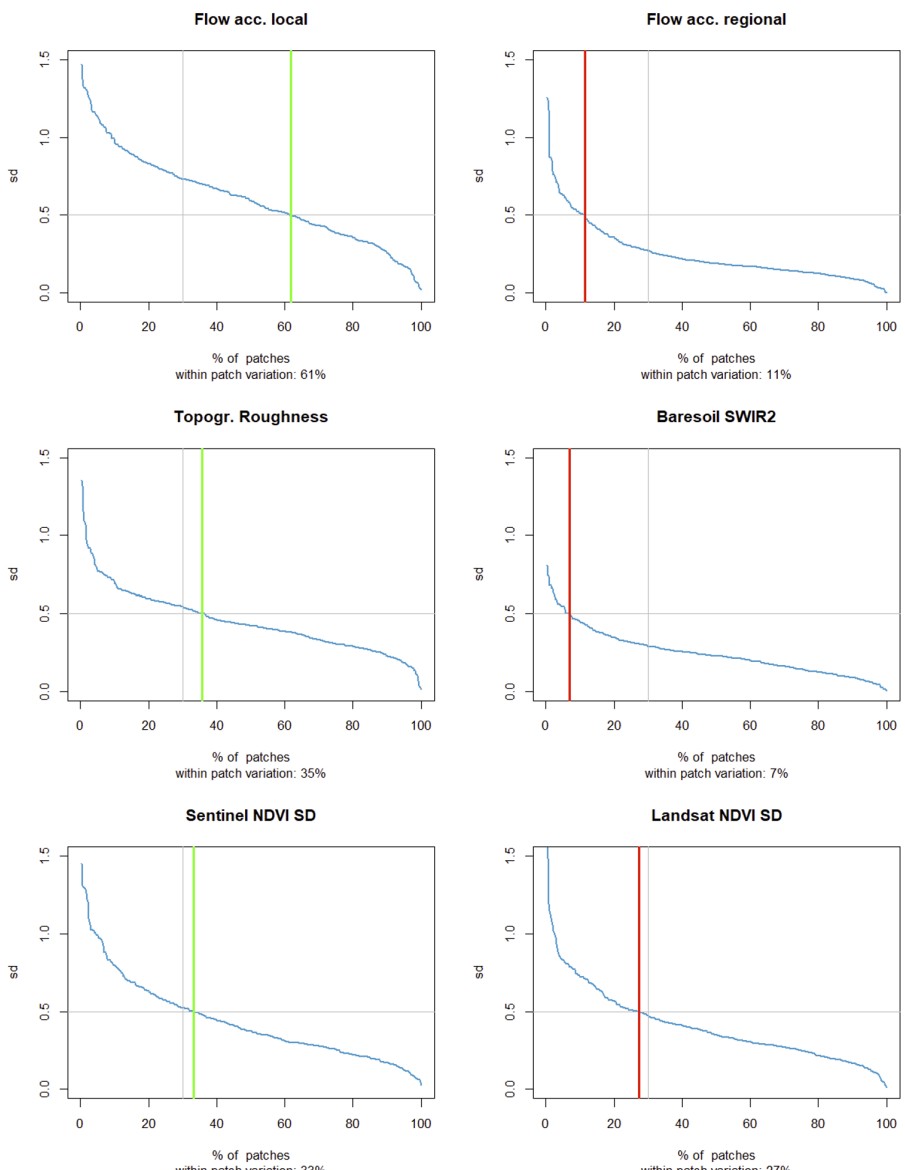

**Figure 7.** Ordered variability (standard deviation) of the potential input data per hexagon. The grey lines show the thresholds, and the green and red lines the percentage of hexagons above or below the threshold.

through the parallel application of the Kennard-Stone and k-means algorithms is employed to identify the locations for the reference measurements, resulting in 90 sample sites for reference analysis .

– the frequency distributions of the reference samples correspond to those of the population (all sampling sites),

– the characteristic feature space of the population is completely covered,



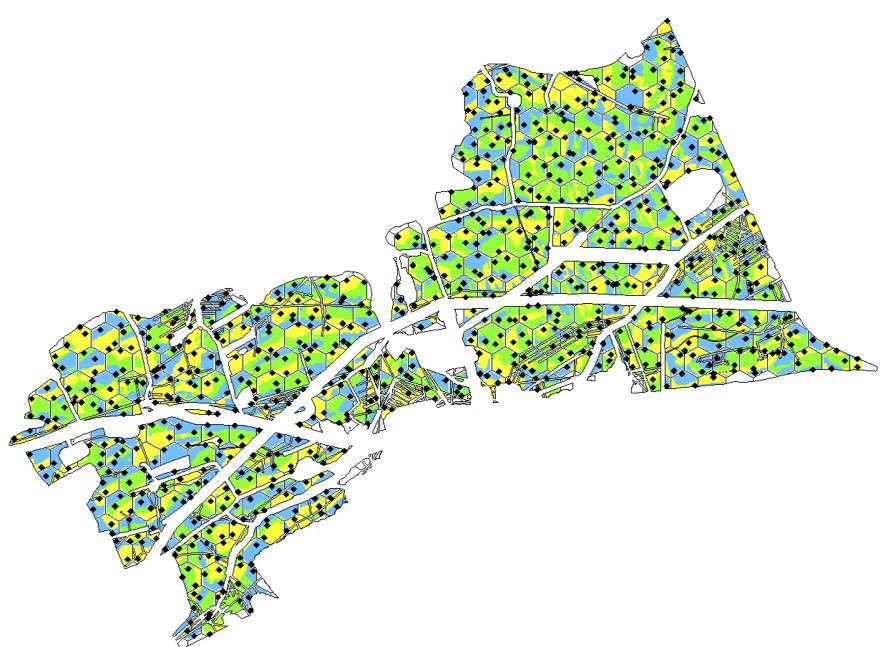

**Figure 8.** Final sampling design. The areas in the background represent the respective alternative areas (maximum of three) where sampling can occur if this is not possible at the designated sampling point.

– the density functions correspond to those of the population, and

– k-means covers all characteristic feature combinations.

The final sampling design of the reference sites at which soil properties were measured using conventional laboratory methods is illustrated in Figure 13. One site was selected from both methods, resulting in a final data set comprising 89 sites.

### 3.4 Spectroscopic modeling

The results of the spectroscopic modeling demonstrate an average explained variance exceeding 90 %. Figure 14 depicts the results for the individual soil properties. The least accurate result was observed for sand content, with an $R^2$ of 0.79. $R^2$ values
exceeding 0.9 were achieved for all other soil properties. These outcomes are likely attributable to the high quality of the laboratory data and the subsequent outlier analysis of the replicate spectra.

### 3.5 Spatial modeling

Figure 15 and Figure 16 show the $R^2$ values of the two models with and without pedotransfer functions, respectively. The models with pedotransfer functions exhibited a 15 % improvement in variance explained for the initial two depth intervals and
a 24 % improvement for the third interval. The pronounced increase in the third depth interval of the pedotransfer model can be attributed to the limited availability of samples in this interval, as the third depth interval could not be drilled in certain cases.





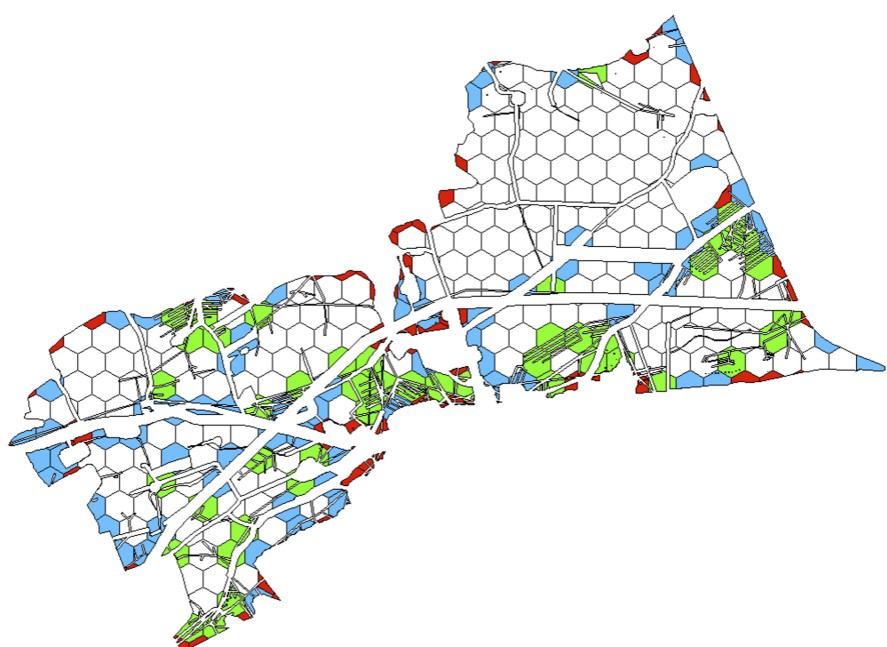

**Figure 9.** Hexagons with specific sampling densities based on their size (white: 3 sample points, mostly complete hexagons; blue: 2 sample points, reduced hexagon area; green: 3 sample points, reduced hexagon area mainly due to drainage; red: no sites due to insufficient area).

The incorporation of pedotransfer models enhanced their stability. Due to the inter-correlation between the soil fractions, the respective results must be treated cautiously. The pH value and the carbonate content have benefited the most, while the soil organic carbon content shows the least improvement. The latter is likely to be the most susceptible to uncertainty due to the influence of management practice and a small local bog, which represents a distinct pedogenetic system and thus introduces a potential bias in the analysis and results.

Figure 17 and Figure 18 show the modeled data.

### 3.6 Scenarios on sampling density

The sampling design approach presented in this paper was developed to enable the systematic reduction of sample density in the context of spatial modeling. This approach permits the construction of models based on three, two, or one sample per hectare while ensuring a uniform spatial stratification. In the case of hexagons comprising solely two samples per se, one sample was randomly selected for density analysis with one sample per hectare.

A reduction from three to two samples per hectare resulted in only slight decreases in predictive accuracy on average for both model runs (see Figures 19 and Figure 21). A reduction in the number of samples from three to one, however, results in a notable decline in predictive accuracy, with a maximum of 25 % of the variance explained (mean = 12 %) for the initial modeling approach without pedotransfer (Figure 20). In contrast, modeling with pedotransfer only results in a maximum reduction of 16 % and an average reduction of 5 % in explained variance (Figure 22).



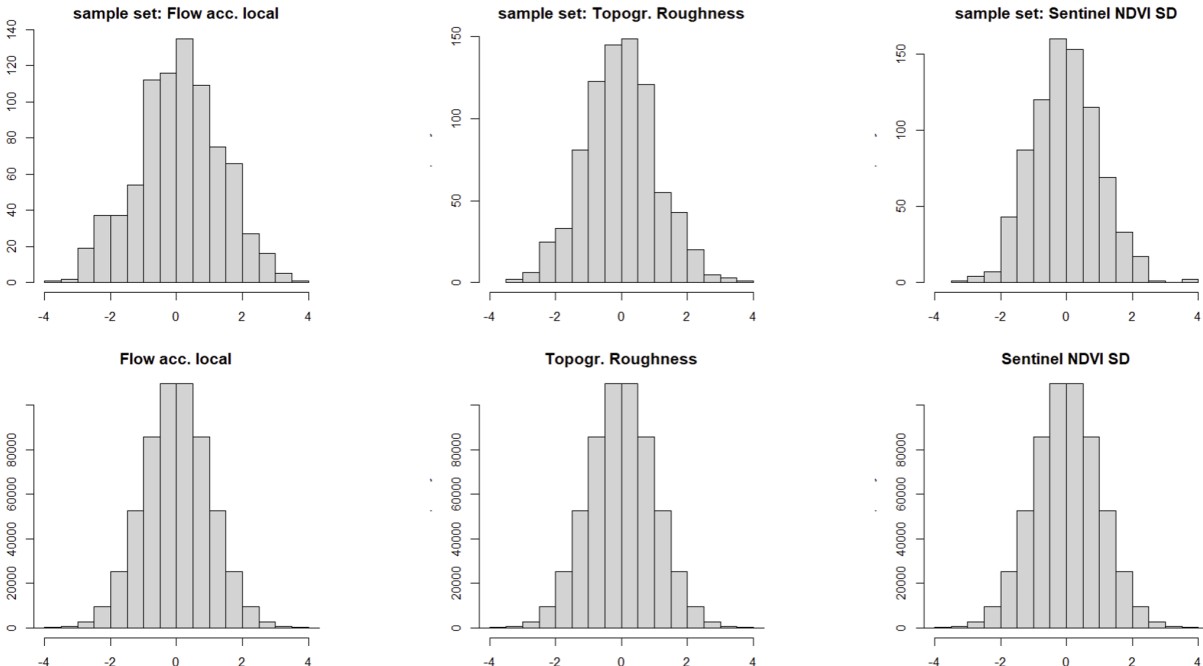

**Figure 10.** Comparison of the frequency distributions of the sample set and the population.

The findings suggest that three samples per hectare are likely unnecessary and that a smaller number of samples would be sufficient for characterizing the spatial distribution of soil in this region. The modeling results indicate that a reduction of one-third of the sampling density would not negatively influence the final modeling results. Furthermore, this reduction could lead to a comparable cost reduction in future projects. The results also demonstrate that the extremes derived from the sequential k-means-Kennard-Stone approach are pertinent and yield consistent predictions across the entire study area. For the modeling with pedotransfer functions, soil organic carbon content was the only variable exhibiting a decline of more than ten per cent $R^2$. The extent to which this can be attributed to non-stationarities and underrepresented soils, such as the small bog, which was not adequately covered by the sampling design, depends on the spatial distribution of the soils. This appears to be the case in this instance. Consequently, a further reduction in sample density may be feasible, depending on the extent of the study area and the distribution and interleaving of pedogenic systems, such as erosion systems, fluvial systems, or bogs. Further investigation is required to substantiate this hypothesis.

## 4 Conclusions

The availability of soil property maps before pedological fieldwork, particularly in the context of soil classification systems that emphasize pedogenetics, represents a significant advancement over the use of classical covariates such as terrain attributes and data on land use. A comprehensive set of basic data on soil attributes from multiple depths reduces the subjectivity of applying



**Figure 11.** Scatter plots, density functions and convex hulls of the samples of the population from the sampling design for the spectroscopic measurements and the reference samples drawn from them using Kennard-Stone and k-means for all combinations of input data.

the surveyor's mental model based on their experience and tacit knowledge. This enables the surveyors to concentrate on their core competencies, namely the recording of the water balance and the pedogenetic description of the soils, which cannot be





**Figure 12.** Comparison of the frequency distributions of the sample set for the spectroscopic measurements and reference sites drawn from them using Kennard-Stone, k-means or an combination of both.

readily discerned or quantified in a laboratory setting. Ultimately, this will enhance the quality and consistency of the soil maps produced.

This study presents a methodology for operationalizing the creation of soil property maps. It relies on a diverse set of pedometric concepts, the integration of spectroscopy, and optimized field and laboratory approaches. Developing an effective sampling design is one of the most critical aspects of operationalizing soil mapping. The concept presented here is focused on the coverage of local variability in a combined geographical and feature space of the input data. Moreover, the data can be analyzed a posteriori to determine whether a lower sampling density would have been sufficient to produce soil property maps with comparable prediction qualities. This can then be adopted to optimize future soil mapping campaigns further and reduce





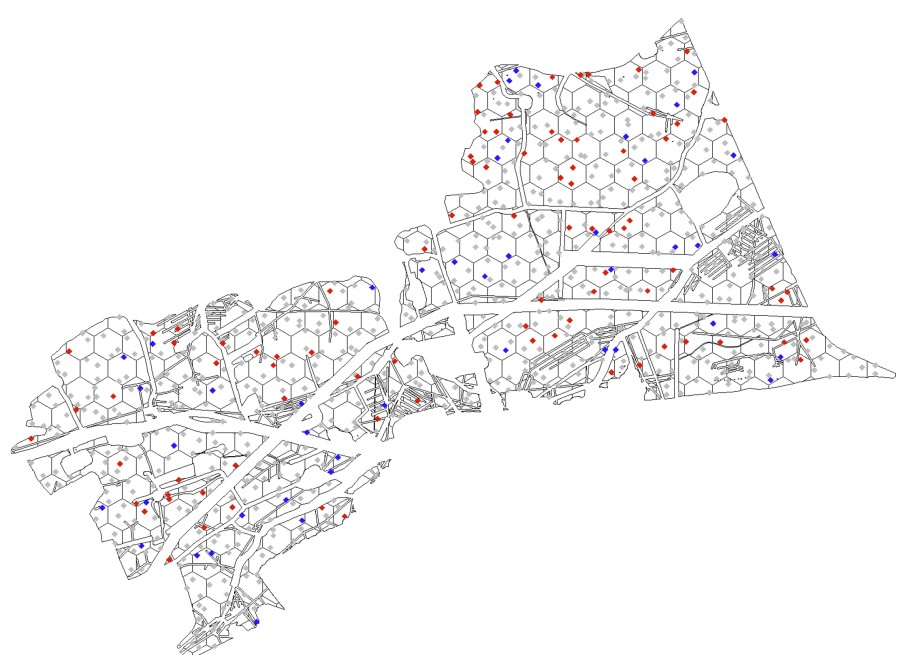

**Figure 13.** Location of the drawn reference sites of the combination of Kennard-Stone (red) and k-means (blue) sites.

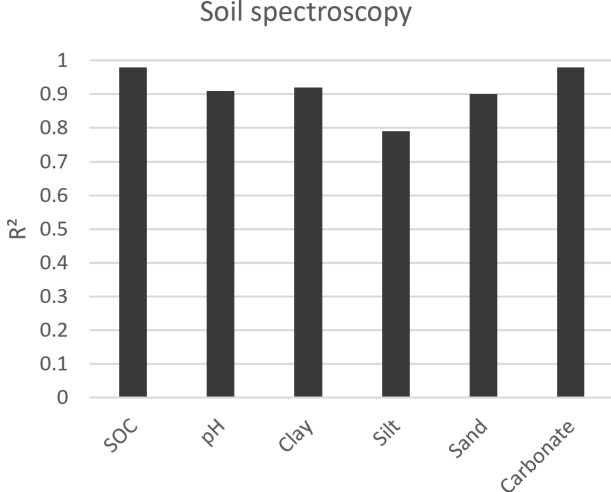

**Figure 14.** $R^2$ values of the spectroscopic modeling results.

costs without compromising quality. The sampling density scenarios demonstrate that at least one-third of the samples can be omitted without substantially reducing predictive accuracy. Consequently, two samples per hectare are sufficient in this area,





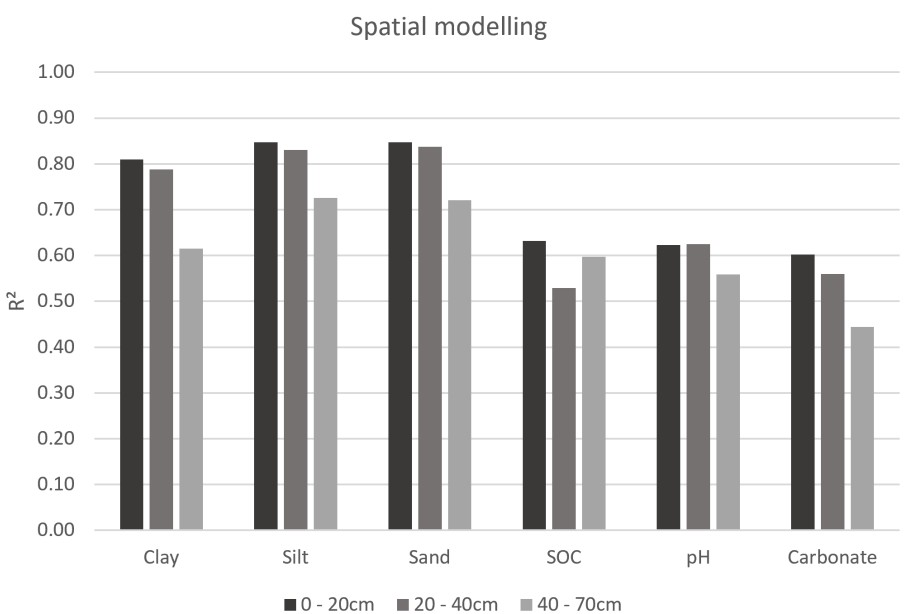

**Figure 15.** $R^2$ values of the initial spatial modeling results.

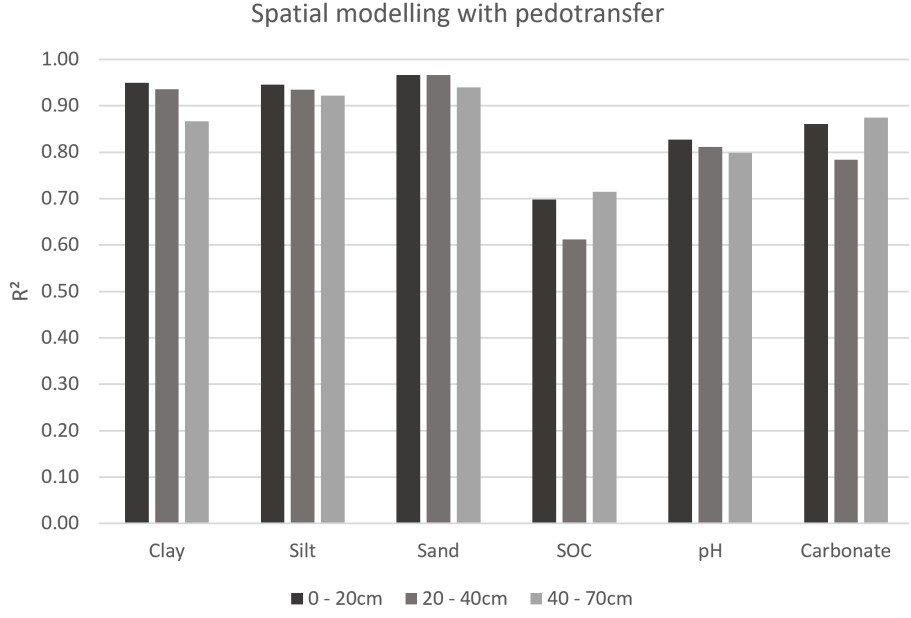

**Figure 16.** $R^2$ values of the pedotransfer spatial modeling results.

345    significantly influencing costs. Future studies must evaluate how this reduction predicts pedological data such as water balance

classes and soil types.




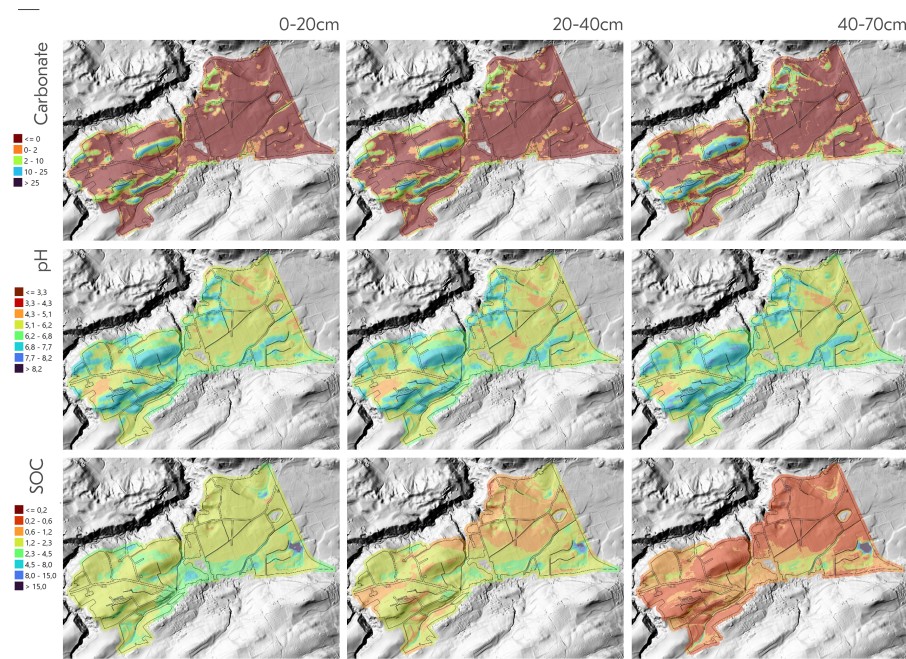

**Figure 17.** Spatial modeling results with pedotransfer for soil organic carbon (SOC), pH-value and carbonate content.

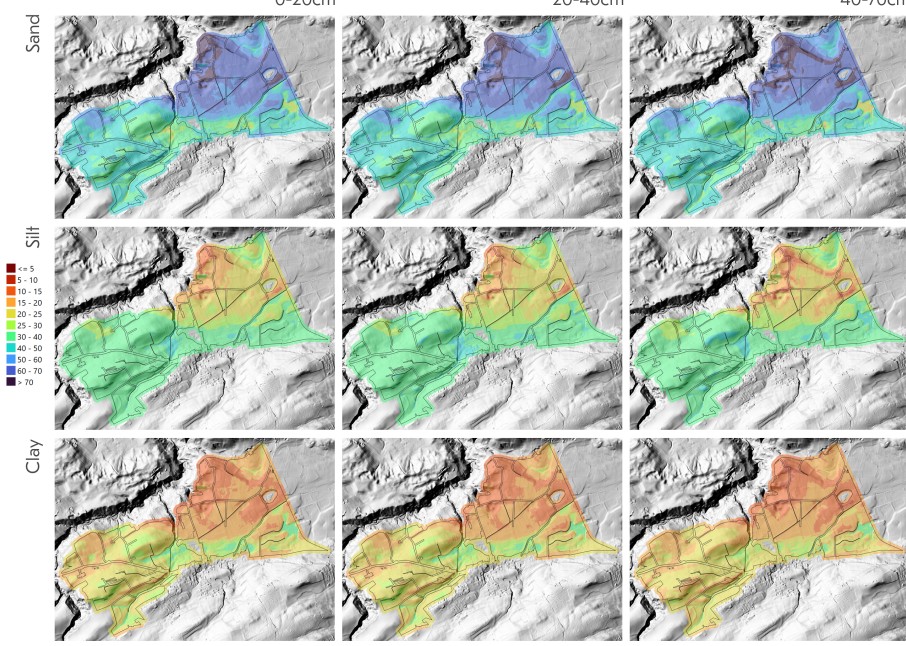

**Figure 18.** Spatial modeling results with pedotransfer for soil texture.




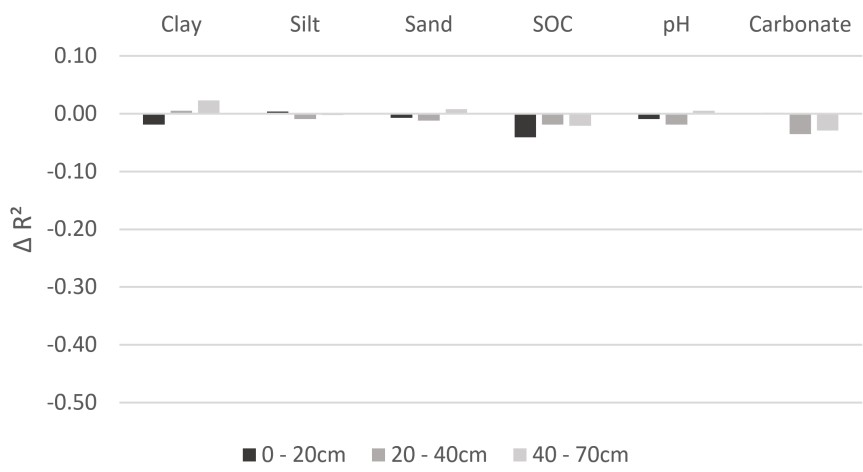

**Figure 19.** Decrease in $R^2$ for the normal modeling approach when the third (average) Kennard-Stone sample is removed from each hexagon.

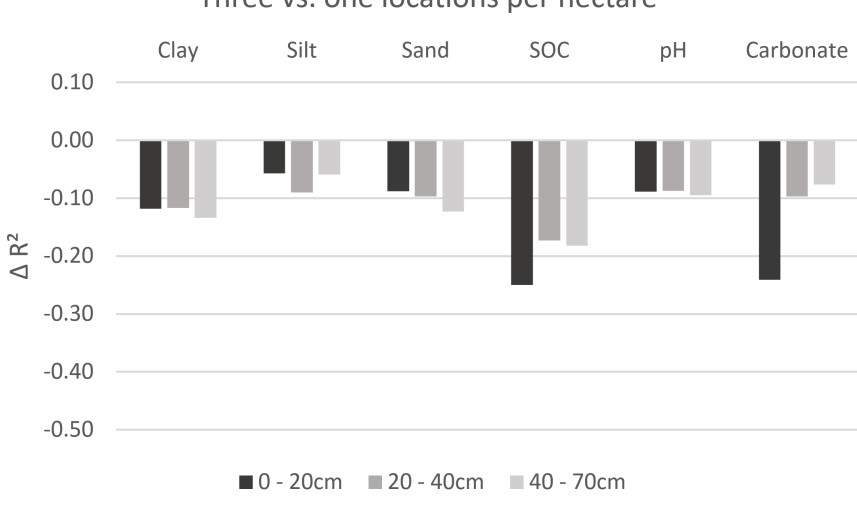

**Figure 20.** Decrease in $R^2$ for the normal modeling approach, when the two first (extreme) Kennard-Stone samples are removed from each hexagon.

The spectral and spatial predictive accuracies achieved in this study are high, demonstrating the potential of combining all approaches in creating soil properties maps and careful sample and data processing. The greatest challenge in soil mapping is transferring techniques into practice, integrating them with traditional field surveys, reducing costs, and increasing the quality of 350 the soil maps. Future projects will aim to integrate the subsequent pedological fieldwork and data science concepts seamlessly.



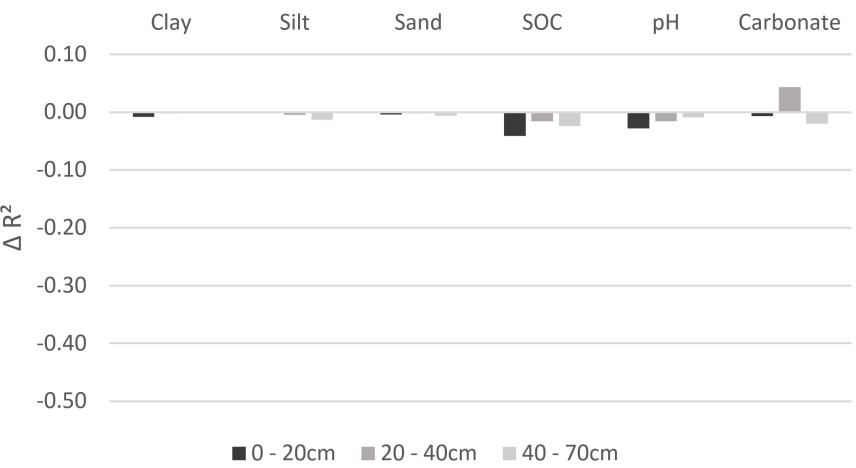

**Figure 21.** Decrease in $R^2$ for the modeling approach with pedotransfer, when the third (average) Kennard-Stone sample is removed from each hexagon.

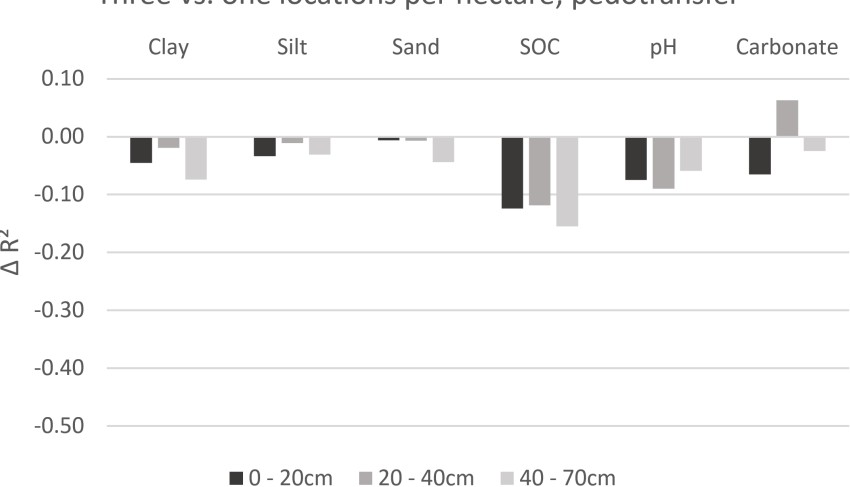

**Figure 22.** Decrease in $R^2$ for the modeling approach with pedotransfer, when the two first (extreme) Kennard-Stone samples are removed from each hexagon.

*Author contributions.* TB conceived the study and developed the new methods. TB and KS performed the numerical analysis. TB and FS performed the feature engineering. TB and RVR drafted the manuscript. All authors provided critical feedback on the concept and methods and contributed to the manuscript.



*Competing interests.* RVR is topic editor of SOIL. Otherwise, the authors declare that they have no competing interests.

*Acknowledgements.* The authors would like to thank Dominik Zahner, Maxime Siegenthaler, and Cwan Kendi of the Swiss Competence Center for Soil for their support during the field and laboratory work.

This research was funded by the Swiss Federal Office for the Environment (FOEN), the Swiss Federal Office for Agriculture (FOAG), and the Swiss Federal Office for Spatial Development (ARE).





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
