# Peer review of "Operationalizing fine-scale soil property mapping with spectroscopy and spatial machine learning"

_EGUsphere, 2024_

## Author Comment (AC2)

**Reviewer 1**

Review EGU Sphere

Behrens et al., 2024
* * *
The direction of the presented work is relevant, and the connection of the different methodological approaches of pedometrics (sampling design, spectroscopy, mapping) is also very important for a practical application. The actual study involved a big amount of sampling and laboratory + spectroscopy analysis.

*Reply:* Thank you!

This study claims to present "a methodology for making operational the creation of soil property maps". However, it mostly presents the application of a complex sampling approach that partly miss the justification of why it needs to be done at multiple stages.

*Reply:* "a methodology for making operational the creation of soil property maps" is not a direct quote. But yes, that is our overarching aim. Operationalization refers to the integration of different methods for a 'real-world' application of where one needs fine resolution digital soil maps. To reduce the costs the sampling density has the be reduced. We do not agree that the sampling design approach is more "complex" than many others. We developed it to be able to systematically draw different subsamples and to explicitly cover the local variability in the feature space within a geographical stratification, which no other design does.

It is not evident why the sampling design need to be at such a high level of complexity. Given the sampling density, maybe even a simple random sample will result to the same accuracy for the maps.

*Reply:* We do not think that this design has "such a high level of complexity". A simple random sample would likely not be useful because it cannot guarantee that we capture the relevant local soil variability with the same number of samples. A stratified simple random sampling might be better than a simple random sample in our case, (e.g. Dick Brus's SPCOSA), however, this also does not ensure coverage of local variability in feature space. Implementing a local coverage approach on top of those approaches would require an even more complex design, compared to the sampling design presented, to systematically and spatially evenly reduce the sample set size to compare different sample densities. This is why we implemented this design.

For example, it miss the integration of the prediction errors of spectroscopy into the next mapping process (a requirement for the claimed framework).

*Reply:* One question is what kind of prediction error(s) or statistics to rely on/use (SE of the stack, bootstrapping, …). An important related aspect is computational demand.

Hence, given the high variances explained, we think this is secondary. Nevertheless, we will try to integrate this in future studies.

Therefore, the manuscript should be re-structured and the introduction need to be expanded by the relevant context mainly targeting the actual presented work. It should maybe just focus on the sampling strategy.

*Reply:* We agree that especially the introduction needs some re-structuring - we will do that. However, we do not see a reason to restructure the manuscript substantially, to focus on the sampling design only. It is clear from the reviewers previous comment that "The direction of the presented work is relevant, and the connection of the different methodological approaches of pedometrics ... is also very important for a practical application".

Moreover, there are probably major problems with the validation strategy (not fully documented, so unclear to know). Cross-validation, how it was likely applied, gives far too optimistics results, and therefore the results are hard to interpret.

*Reply:* "The evaluation of all models was conducted using 5 times 10-fold cross-validation." We used a common 10-fold cross-validation approach as implemented in the R package caret. We repeated this 5 times to achieve stable results. There is some debate on cross-validation of spatial data with respect to autocorrelation. In a recent paper Wadoux et al. (https://doi.org/10.1016/j.ecolmodel.2021.109692) conclude that cross-validation is only problematic if the samples are clustered, which is definitely not the case in the data presented in our work. One aim in developing the sampling design was to avoid spatial clustering of the sample. We are wondering why a cross-validation should give "far too optimistics results" and compared to which other approaches? Leave-one-out can produce overoptimistic results, but we did not use LOO-CV. If you refer to nested CV, the differences are usually very small.

In some parts of the manuscript, relevant informations are missing. Please accept the subsequent detailed comments to support my claims above:

Abstract:
* * *
L3: Authors claim a novel approach to soil mapping. Sampling design seems to be new, but the rest are established methods.

*Reply:* We wrote: "As part of a novel approach to soil mapping, we integrate...". This is part of the context on why we are working on operationalization. The novelty of our

approach, as stated in the manuscript, is not the soil mapping itself, but the integration of modern pedometric methods in an operational 'real-world' federal soil survey.

L6: Subjectivity of soil mapping, rather "field soil description"? Soil mapping by pedometrics methods as proposed by the current article should already have reduced subjectivity.

*Reply:* Thanks, we will replace "soil mapping" with "field soil description"

L24: Soil maps are available on coarse scale (e.g. european or global maps, national maps?), but their information content and/or resolution/scale is not sufficient.

*Reply:* Thanks, we will add "fine or medium scale" to make it clear.

Introduction
* * *
The introduction is very poorly structured, i.e. every paragraph provides a new objective of the present study that was not well introduced in the first section of each paragraph.

*Reply:* We will restructure the introduction.

Moreover, the line of thought is not well supported by existing research on the subject nor well argued for. Some examples:

First paragraph: the authors detail parts of the mental model used in conventional survey and how it can be supported by soil property maps. It remains unclear when it comes to the role of the mental model in todays digital soil mapping approaches. The process of Gestalt shift and how it can be supported by the proposed method does not become completely clear from the description.

*Reply:* In pure digital soil mapping approaches the role of the mental model surely does not play any role. However, we are aiming at integrating digital soil mapping with traditional field surveys. We mentioned this in the first sentence of the abstract as well as in L27/28 ("to generate soil property maps for soil surveyors to use in their pedological fieldwork."). We will improve the introduction to clarify our meaning.

In L45 reference scales are mentioned, however, the study then presents digital soil mapping approach having a pixel final resolution (unclear, likely 2m as the predictors were prepared at 2m). It is not introduced which assumptions are often made regarding scale and point density in conventional surveys (see e.g. Legros, La cartographie de sols) which might be relevant as the study compares different point densities.

*Reply:* We reported this in the following sentences "four locations per hectare" aiming at a scale of 1:5.000 (Siegrist and Marugg, 2023; AfU Solothurn, 2024, nonetheless, as stated above, we will improve the introduction to clarify our meaning.

L36-37: It stays unclear that or why end-users need a higher density of analytical data. Also, that it improves the quality of thematic maps. No argument or citation/evidence is provided.

*Reply:* For fine resolution digital soil mapping one needs well sampled data at an appropriate density to capture soil property variability, which can be high over short distances. This of course is by now well known.

L36: What is exactly meant with thematic maps? Soil ecosystem services?

*Reply:* We will add some examples: maps for spatial planning, flood protection, natural hazards, agriculture, nature conservation, and climate adaptation.

L46: Do you maybe mean soil wetness/waterlogging instead of soil moisture? The link with soil quality is likely weaker with the latter (depending on definition).

*Reply:* Thanks, yes "soil wetness/waterlogging" are better terms.

L50: What is with the 3 remaining observations, are they not recoreded by a surveyor?

*Reply:* Yes, they are used to building their mental model.

L52: This study is not the first to investigate the relationship between sampling density and predictive accuracy, however, no link to findings of other studies is drawn at all (see e.g. Kempen et al., 2014).

*Reply:* We could not find a paper from Kempen et al., 2014. For sure there are other studies. Many of them focus on large area (> 100km$^2$) or rather small ones at the field scale (see Schmidt et al., 2014). We have not claimed that our study is "the first to investigate the relationship between sampling density and predictive accuracy". Since the main focus of this paper is based on a specific scale and sample density, it is just "one objective of the operationalization project" (L51).

L55: Brus, 2022, references a hole text book. It remains unclear if this reference is supporting the whole sentence or just the mentioned methods. Please give at least a chapter or section to make it clear.

*Reply:* We thought this is clear, since we mention "geographical stratification or spatial coverage". We will provide the chapters.

L55ff: Either the reasoning needs to be more detailed, providing evidences for the statements in the text, or it needs to be supported by the literature. Does the overrepresentation of the small areas depends from the configuration of the study area or from the variables chosen for the sampling design?

*Reply:* It makes no difference. For sure the "variables chosen for the sampling design" should reflect "the configuration of the study area".

L59: k-means and Kennard Stones are not sampling designs, they can be used to create them.

*Reply:* True for k-means, which is commonly used for stratification in a sampling design. Kennard Stone is an algorithm to sample a calibration set that is representative of feature space. To prevent any confusion from the readers, we will rephrase the sentence.

L62: .. not relevant in most cases. ... To address both of these issues.. : I have difficulties to identify two issues, please clarify. Moreover, if the issues are not relevant, why are they address at all?

*Reply:* Thanks, the formulation including "not relevant" is not precise. We will rephrase this sentence. Issue 1, L56: "regions that exhibit variability receive more samples"; issue 2, L59, "In addition, sampling designs ... tend to identify new transition zones."

Overall introduction: strong focus on sampling design, however, the study shows many other aspects as well. As clear, concise objectives are missing, it remains unclear what the authors truly want to present. Or, if the reader is just dumped with a large number of used methods combined around a sampling design.

*Reply:* We will restructure the introduction. We present a sampling design, yes, but the main case is this combination of carefully selected methods, the accuracies achieved with them, and the analysis of the sampling densities. All tailored towards the operationalization of an integrated fine scale (traditional and digital) soil mapping approach.

Methods
* * *
Section 2.1: It remains unclear what soil types are to be expected in the study area. There is no information on climate, geology or geomorphological processes. For the transferability, i.e. the limitations to a specific study area, such background information is very relevant. It remains therefore unclear, if there is geological variation within the study area that has been neglected.

*Reply:* Transferability of what? We are aiming for a method that's generally applicable for mapping soil properties at that "scale". Although we felt that detailed information on the study aera was not necessary for the research presented in this paper, we now see that some information on the location will help readers. We will add that information.

Moreover, sampling has been done by fixed depth intervals. Neglecting genetic soil horizons may be done, but not for soil types that have small horizons with abrubt or large changes in properties (e.g. diagostic horizons in podzols).

*Reply:* The sampling has been done by fixed depth intervals, because the aim is to generate consistent information for larger areas. We either generate soil property maps for the pedologists before they start their field work, which is our aim here, or we generate soil property maps after the pedological field work, but then based on subjective descriptions and open questions on how to build spatial soil property maps based on (diagnostic) horizons across different pedogenetic systems. In real-world surveys once cannot have both.

L88: What are exclusion areas? Will those be mapped, but not sampled?

*Reply:* (L78): Sampling and mapping "roads, drains, ... and residential areas" makes no sense, if we want to generate soil property maps. We map but do not sample areas of gas pipes and electrical wiring, when covered with soil.

L95: What were the five different settings?

*Reply:* We will provide some future explanations in the revision. Note that this is the topic of a separate paper that is currently being submitted. We hope to cite it before this paper is published.

*Reply:* L115: It seems very strange that bare soil reflectance can be extrapolated to permanent grasslands. But, this seems published work from a co-author.

Yes, according to this study it is especially helpful for grassland. Also, many grasslands were once ploughed and if the feature space is similar, it makes sense.

*Reply:* L117: The resolution of the landsat derived data was changed by "spatial modelling with machine learning". Please add details how this was done. It does not seem a default method.

*Reply:* It is the same method used for spatial soil mapping of the soil properties. We will reference it.

L120: How was the selection of the predictors made? "carefully" does not inform about the approach. How was the de-correlation approached?

*Reply:* The selection was based on expert knowledge on the basis of the feature importance analysis from previous studies. Another criteria was a correlation below r = 0.7 between the datasets. We will add this information to the revision.

L128: Is this a rank transformation? If yes, maybe mention it to make it easier to understand for the readers.

*Reply:* Yes, it is. We will add that detail.

Section 2.3.1: Using hexagons has the mentioned advantages, but, in the given study area, the area to be sampled is irregular as there are streets removed from the hexagons. The reduction of sampling points seems somewhat arbitrary. Would a clustering by spatial coordiantes proposed in Brus, 2022, not yield better distribution of spatial sub-areas?

*Reply:* The problems with streets etc. would be the same. We reduce the sample density if there are streets and buildings in the hexagon but not if there are drainages. There might be some advantages at the boundaries when using clustering by spatial coordinates, but this is not relevant for our approach. The advantage is that the hexagons are evenly distributed and of same size.

L67: It remains unclear how n and p where determined and what would be the rationale behind it.

*Reply:* (L167): Yes, we forgot to provide this information: n and p were set to 2 and 3. The rationale is given in L168-172.

L190: How were alternative areas defined, size? Why not alterantive sampling points?

*Reply:* "The Euclidean distance in the feature space was the basis for creating the alternative areas" (L192-193). At the time the sampling design was created, it was impossible to predict when which area can be accessed (vegetation, wild bulls). Therefore, countless alternative locations would have to be generated/selected. Providing alternative areas is therefore a much more practical approach in the context of operationalization.

Section 2.3.2: Where these samples taken from the original sample set of 812 or were these another new sample of additinal 45?

*Reply:* "a subset of the samples" (L195)

L212: Using grinded soil samples for the subsequent analysis is very unusual, what is the justification for that? And maybe indicate how fine grain was the grinding done?

*Reply:* Grinding (<100nm) is required and only applied for the MIR measurements. We will clarify this.

L214: texture by sedimentation, do you mean the pipette method? Please give a reference, also for SOC and carbonates.

*Reply:* We will provide more details on those standard methods.

L218: Were the replicates removed based on Euclidian distance between the replicates or distances computed from within one spectral response?

*Reply:* Based on Euclidian distance between the replicates. We will rephrase this to be more precise.

Section 2.5.1: It remains unclear what hyperparamteres were tuned and how (what candidate values and what procedure to select them, likely cross-validation).

*Reply:* "The hyperparameters of the models were optimised using the R packages caret (Kuhn and Max, 2008) and caretEnsemble (Deane-Mayer and Knowles,2023). The evaluation of all models was conducted using 5 times 10-fold cross-validation." Caret applies a grid search on predefined settings to tune the hyper-parameters. We will include some more details in the revision. In the context of this paper, the specific parameters that were tuned are of limited relevance, given that the focus is not on a direct comparison between different models and settings.

L232: Most likely, the model performance results are too optimistic. According to this section 5 times 10fold cross-validation was applied. Since no further mentioning, splitting was probably done at random ignoring the fact that the samples from different soil depth are not independent observations. Cross-validation would need to be done at least by a leave full locations out splitting. Moreover, it remains unclear, how the model tuning was done and especially the stacking. Most likely selection of model predictors and model paramters involved using the cross-validation sets, repeatedly. Therefore, the final reported cross-validation error metrics are not indepenedent anymore from the fitted model and are too optimistic compared to only one single run of cross-validation and certainly compared to an independent randomly sampled data set (e.g. Brus 2011). Moreover, the "pedotransfer" approach (see next comment) does also strongly confound the cross-validation as most likely maps were used produced with data that was left out for "independent" validation. Reported cross-validation results for the final maps are therefore likely far too optimistics.

*Reply:* Spatial modelling was conducted separately for each depth interval. There is some debate on cross-validation of spatial data with respect to autocorrelation. A recent paper of Wadoux et al. (https://doi.org/10.1016/j.ecolmodel.2021.109692) concludes that cross-validation is only problematic if the samples are clustered, which is definitely not the case in the data presented in our work. Based on the experience from previous projects, one aim in developing the sampling design was to avoid clustering. Hence, in this respect the reported cross-validation results for the final maps are likely not too optimistic.

Section 2.5.2 in general: It remains unclear if and how uncertainty was quantified for the stacking approach. Moreover, it also remains unclear, how previous models were included as a pedo-transfer function. Were maps created from all soil properties and then in a second step those maps were used as predictors for another model fit? This seems rather unusual and should be clearly explained and also the improvement transparently discussed in the results

*Reply:* As described the stacking approach was validated using 5 times 10fold cross-validation. Yes, the maps were included as predictors (L234-235). We also transparently

discuss the improvement: "Due to the inter-correlation between the soil fractions, the respective results must be treated cautiously." (L303-304).

(currently it is not clear if "pedotransfer function" in results and plots refer to the spectral transfer functions or this assumed appraoch).

*Reply:* Section 2.5.2, refers to "Spatial modelling".

Section 2.5.3: Maybe I overlooked it, but for the mapping scenarios with a reduced number of sampling locations it remains unclear, how the validation was done. Was there cross-validation applied to the data used for training? This would then mean that the CV sets are not all the same for the different scenarios and that there is a maybe considerable variation to be expected only due to the different sets. As the variation of the 5 times repeated CV is not shown, it is difficult to estimate the variability of different, i.e. also smaller CV sets.

*Reply:* The same validation procedure was used and for sure the sample set size is reduced. Since the accuracy decreases the variability is probably higher, but that does not change the result or the meaning of the results. To achieve stable results, we use 5 times 10-fold CV.

Model evaluation overall: It is not clear how R2 was computed, was it computed as the MEC model efficiency coefficient which use is widespread now. In addition, the predictions could be more biased for low sampling density, this is not computed/presented.

*Reply:* As written in the paper we calculated the $R^2$ not the Nash–Sutcliffe model efficiency coefficient. All measures have their limitations. Instead of the MEC, we would have used the Kling–Gupta efficiency or Lin's Concordance Correlation Coefficient. The $R^2$ is employed as it is likely to be the most prevalent measure and therefore the most readily comprehensible to those utilising the maps generated. Since the $R^2$ is lower for lower sampling densities, a higher bias and/or variance are expected.

Results
* * *
L260ff: Background on imbalanced data situations, should rather make part of the introductions. Instead, here a proper discussion of the findings would be better.

*Reply:* This was not an aim, but rather a side effect. Hence, we would not focus on it in the introduction.

Figures 14ff: Barplots displaying R2. Report how R2 was obtained in the figure caption, is it a mean or a median of the 5 repeated cross-validation runs? Moroever, barplot are not suitable to display the results, because for a 5 times repeated cross-validation a stripplot showing the variability would be more suitable. If only one R2 value per response is shown as in figure 14, a table might be more suitable as the information desnity of the graph is only minimal given the space it takes up in the article.

*Reply:* As usual it's the average. We will convert it into a table.

Section 3.4: Spectroscopy models have non-neglectable errors. How were those considered, if at all, in the subsequent analysis?

*Reply:* The spectroscopic models show high accuracies. We assume that errors due to field sampling, uncertainties in locations, etc. are higher. This must be seen in relation to the purpose of the mapping as well as final mapping scale. Local variability in the soil is high. Every soil map is generalized to some degree. This inevitably happens here as well. However, we agree that in future studies some error propagation should be included. This study focusses more on the general operationalization aspects, especially since all models show high accuracies.

Figure 11, 12: There is a lot of information shown at left for interpretation to the reader. Those are the only arguments why the sampling should outperform a simpler one. There is not enough prepared evicence presented. I am not sure if the distribution argument holds (the more similar the distribution of population and sampled location, the better the R2 of the mapping), at least in the introduction does not give evicence that this relationship is strong enough to justify the complex approach.

*Reply:* We disagree that the approach we took is complex. All we show in these figures is that the sample set drawn is representative of the population, which we think is important. That is all.

Further comments:
* * *
Overall, there are too many figures. Please evaluate if they are truly all needed, some information could be combined into one figure (e.g. for the barplots).

*Reply:* Thanks, we will follow your advice.

L84: For datasets use citation that also appear in the reference list.

*Reply:* We followed the terms of use for this dataset and provided the correct reference in the text:

https://www.swisstopo.admin.ch/de/nutzungsbedingungen-kostenlose-geodaten-und-geodienste

L220: use uppercase title.

*Reply:* Thanks!

Figure 1: It remains unclear, what the colors mean. Not all steps are quite clear, there is a lack of detail in the figure, i.e. field work is completely missing.

*Reply:* We will provide information on the meaning of the colors. This figure as written in the caption in an "Overview of the data and processing steps." This does not comprise field work.

Figures in general: color scales are not color blind friendly.

*Reply:* They should work fine in black and white

Figure 14: The information content does not justify a figure of this size.

*Reply:* We will provide a table instead.

Figure 17ff: Legends are too small. The units are missing.

*Reply:* We will update these figures.

Figure 19ff: Use figure captions instead of figure titles (that are partly incomplete, i.e. what is the meaning of "…; pedotransfer".

*Reply:* OK.

---

## Author Comment (AC3)

**Reviewer 2**

The paper introduces a framework for high-resolution soil mapping of various properties at small to medium scales, with a small emphasis on cost-efficiency. Its novelty and relevance come from the combination of multiple Digital Soil Mapping (DSM) techniques in a practical context. The authors propose a new sampling design, employ diverse feature engineering methods for remote sensing data, and utilize a "two-step modeling approach."

*Reply:* The reviewer is mostly correct, but we did not mention a "two-step modeling approach." We combine various pedometric methods for operationalizing soil mapping.

In this approach, they initially use spectroscopy to generate additional training data for the final spatial model based on remote sensing data.

*Reply:* This is correct.

Their pipeline incorporates various state-of-the-art methods.

*Reply:* This is correct.

However, the paper is not always easy to follow because the authors introduce numerous topics and research goals within a single paper.

*Reply:* The paper deals with various methods and that this is one novelty and point of the paper - the operationalisation of modern pedometric methods in a real-world soil survey. Actually, there are only two research goals here. 1.) Generating high resolution soil maps as good as possible (integrating spectroscopy and spatial machine learning) and 2.) a sampling design, which allows for constrained subsampling to test the number of samples required for future studies. We will clarify this in the revision.

This led to the drawback that specific important aspects for a functional framework were addressed inadequately, whereas other topics got way too much attention:

The primary aim of the paper, as I understand it, is to present a state-of-the-art framework for DSM modeling that leverages the combination of various DSM methods. Hence, focusing so intensively in the introduction and abstract on how soil surveying could benefit from DSM products seems rather irrelevant to the actual scope of the paper, since "the value of DSM maps for soil surveying" is not addressed in the Methodology or Results & Discussion section anymore.

*Reply:* This is true that the value of DSM maps for soil surveying "is not addressed in the Methodology or Results & Discussion". However, it is relevant in terms of context and for the scope of the paper. It provides the context for our aims and methodology. We will revise and restructure the introduction to ensure that this context is clear with relation to the research presented.

Then the paper introduces a new complicated sampling design as part of the framework.

*Reply:* The sampling design is no more complicated than other existing designs that use stratification eg. SPCOSA or LHS. With our approach we i) derive hexagons, ii) draw a k-means based sample sets, iii) draw a KenStone sample sets. That is it with the advantage that our method helps to cover local variability in feature space and allows for systematic sub-sampling.

While a long theoretical rationality behind the sampling design was provided, no real evidence was shown that this sampling design is actual capable of improving predictions.

*Reply:* The aim was a sampling design, that covers the local fine scale variability and to be able to systematically subsample from that design to find a sample set size required. The main aim was not to improve predictions.

Generally, no evidence was shown that this long pipeline used in this paper was in any way more appropriate than a more simplistic approach.

*Reply:* What is a 'long pipeline'? Sampling design, spectroscopy, spatial modelling, validation and mapping? To us, this is what one needs to do to operationalise DSM. What does the reviewer mean by 'more appropriate'? Than what? Than traditional soil mapping? And what would be more simplistic than the approach we took? No spectroscopy? Only simple random sampling? Perhaps no covariates and only Kriging? Our aim was to integrate modern pedometric methods to operationalise DSM. Our aim was not to compare approaches. Our aim was to try to generate the best maps possible out of the box. Hence, we systematically combined approaches, which proved promising and helpful in the literature.

On top the researchers also wanted to address the questions of how the sample size may influence prediction performances.

*Reply:* Sample size is one of the (if not the) most important questions in operationalizing soil mapping and why we presented that here.

On the other hand, information on other aspects like modelling is minimal, which even raises concerns on data-leakages.

*Reply:* All the modelling methods applied have been previously published. We do not understand what the reviewer means by 'data-leakages' and to which degree do you expect it has an influence on the models and the accuracies?

The stretch of proposing a new sampling design, while discussing the importance of digital soil maps for soil surveyors harmed the original goal of presenting a functional framework.

*Reply:* The aim of our paper is the integration and operationalization of modern pedometric methods in a soil survey. "The importance of digital soil maps for soil surveyors" is a key idea to make pedological work more efficient. This is the context

behind this study, which we provided in the introduction. The sampling design is an integral part of increasing the efficiency of soil surveys. The approach presented is a functional approach for a modern soil survey that uses the latest pedometric techniques.

I propose that the revision should entirely focus on either (1) presenting a clear framework that allows to reproduce their combination of methods, potentially with code, and less focus on e.g., the sampling design and soil surveying,

*Reply:* As described above, our goal is to integrate modern pedometics methos and operationalize them in a 'real-world' soil survey. The combination of methods was used as building blocks to generate accurate soil property maps that can be used to inform a soil survey. We will revise the paper to make this point clearer, but we disagree that we should change the focus of our paper.

(2) discussing how and why DSM maps are relevant for Soil Surveyors

*Reply:* We did that in the introduction to provide context for the work presented in the paper.

or (3) showing the advantages of their new sampling using a benchmark compared to other sampling designs.

*Reply:* A comparison of soil sampling methods is not the aim here. Our design fits our purpose: to systematically sample and cover local variability. Benchmarking would make out paper longer and would distract from our acutal aims.

However, I suspect that the latter is not really possible without a second sampling campaign, and the first would be most interesting given that the novelty comes from combining so many different methods.

*Reply:* Sure, this is kind of novel. However, it is not the driver of this work. We will work on this as well on future projects and hopefully come up with a synthesis of what is really required and when. In this publication we want to introduce the general approaches and ideas and how we combined them. It was challenging to put this into practice as well as into one paper.

Please see below specific comments:
* * *
**Abstract**

**L. 2 – 3:** *"The latter is paramount, as they [Soil Maps] form the basis for many thematic maps."*

The authors probably want to say that soil surveyors can use DSM maps to create new "thematic maps". This only becomes clear after reading the introduction or follow up sentences in which they introduce the concept of soil surveyors using DSM maps. When first reading the abstract, it is not clear what is meant by thematic maps.

*Reply:* We will provide examples.

**L. 10 – 11:** *"Methods to reduce the uncertainties inherent to the spectral and spatial data were integrated."*

This seems too vague and could mean everything, because "uncertainty" has a lot of context-specific definitions. Given, that this was only a small part of the actual methodology, this sentence may be dropped.

*Reply:* It is the abstract where most methods can't be explained in detail.

**L. 19 – 20:** *"Our study highlights the value of integrating robust pedometric technologies in soil surveys."*

The authors did not really give evidence for this (e.g., they did not show how integrating their framework improved soil surveys).

*Reply:* Yes, this is a bit misleading. We will drop that sentence.

Rather the value of this study comes (or should come) from presenting a functional framework in which various pedometric technologies are effectively combined.

*Reply:* We hope it comes, although this is not our main goal.

**Introduction**

**L. 21 – 33 & L. 40 – 45:** In the introduction, the authors extensively discuss how soil surveyors could benefit from DSM products. However, it is unclear how this relates to the paper's primary goal of presenting a framework for DSM modeling. This section should be much shorter, as I, as a reader, expected a paper that integrates the soil surveying aspect within the framework. Yet, the actual paper just focuses on the DSM modelling, which is detached from the introduction. I understand that this work was conducted within a project where the goal is to create DSM maps for soil surveyors but this is not really relevant for a general framework on high-resolution DSM.

*Reply:* We agree that this is a bit misleading. However, it is the context which we should provide. Please also see our reply above.

**L. 48 – 50:** Four samples per hectare sounds like a lot. Is this common- or best practice in Switzerland? Maybe a citation could help to clarify this.

*Reply:* We provided a citation: (Siegrist and Marugg, 2023; AfU Solothurn, 2024).

**L. 47 & L. 53 – 69:** The authors' main argument is that a targeted sampling design (i.e., a sampling design that covers the feature space) does not provide even geographical stratification. However, it is difficult to understand why the authors put such a great focus on the spatial coverage and the concept of local extremes without any reference that supports their line of argumentation. Spatial coverage might not even be associated with an increase in performance for DSM modelling as for example indicated in Wadoux et al. (2019).

*Reply:* We are very aware of this. We focus on this because of being able to systematically subsample. Moreover, it is not only about spatial coverage. The design explicitly covers local variability of the feature space, which no other design does. We also show that the feature space of all covariates is covered, even for the ones with lower spatial frequency.

Conversely, it has been repeatedly demonstrated that feature coverage can enhance predictive power, at least when compared to Simple Random Sampling (see, for example, the discussion in Žížala et al. 2024). The cited work by Brus (2022) also refers to this concept at the beginning of chapter 18 and the end of chapter 19.

*Reply:* Sure, and as mentioned above and written in the paper, feature space coverage is included in the design presented. We also show that the feature space of all covariates is covered, even for the ones with larger spatial frequency compared to the ones used within the hexagons for covering the local variability. So, the feature space is covered, although only locally optimized. Figure 10 shows that the margins of the frequency distributions are slightly oversampled, which should be helpful for the ML algorithms (see research on imbalanced data).

Finally, spatial coordinates can be incorporated into targeted sampling to increase spatial coverage if desired.

*Reply:* Sure, and we are aware of this. But this does still not allow for the systematic subsampling we applied in this study and might still result in spatial clusters. Once we know how many samples are required (after multiple further projects) we can then switch to another design.

**Wadoux, A. M. C., Brus, D. J., & Heuvelink, G. B. (2019).** Sampling design optimization for soil mapping with random forest. Geoderma, 355, 113913.

**Brus, D. J. (2022).** Spatial sampling with R. Chapman and Hall/CRC.

**Žížala, D., Princ, T., Skála, J., Juřicová, A., Lukas, V., Bohovic, R., Zádorová, T., & Minařík, R. (2024).** Soil sampling design matters - Enhancing the efficiency of digital soil mapping at the field scale. Geoderma Regional, 39, e00874

**Methods**

When introducing a framework, the ultimate goal is for other researchers or DSM practitioners to be able to reproduce the methodology for future DSM campaigns. However, the absence of provided code is a significant drawback. Given the numerous methods employed for feature engineering and the complexity of the sampling design, it would be challenging to reproduce any of the pre-processing steps without access to the code.

*Reply:* All methods are previously published and standard pedometric techniques that are not particularly complex. Compared to other methods the sampling design is also not complex – it is based on two exiting common methods.

**Section 2.2.1:** A wide range of features have been used and engineered. To better organize and track these different features, an overview table would be beneficial. This table could include columns such as the type of feature data (e.g., DEM, terrain attributes, bare-soil multispectral RS, etc.), the engineering/processing applied (e.g., multi-scale), and the dimensionality of the features as a numerical value.

*Reply:* We do not think that a list with 600 names of parameters is useful here. For example, a Gaussian pyramid is an overcomplete representation. We have even added intermediate levels for modeling purposes. It is not relevant here whether one or two scales or levels more are included. It's more about the overall approach.

**L. 114 – 115:** This is not clear even with the reference. Was the bare-soil multispectral data predicted given the other available features for these affected areas?

*Reply:* Yes, but only for grassland. We will clarify this.

**L. 120 – 121:** There are several questions that should be addressed by the reviewer about the selected covariates for the sampling design:

(1) It is stated that "a combination of carefully selected uncorrelated covariates" were used for the local feature coverage of the new sampling design. Afterwards some rationality behind the picked features is given. This implies that features were handpicked based on expert knowledge and intercorrelation and not picked based on e.g. an automated correlation matrix filter. It would be better to be more explicit about this and make clear that they were handpicked.

*Reply:* The selection was based on expert knowledge on the basis of the feature importance analysis from previous studies. We will add this information to the manuscript.

(2) Why were Sentinel 2 and Landsat NDVI SD selected? Although it is mentioned that they are based on different time intervals, they appear to be strongly correlated in Fig. 3. As a result, NDVI SD will be heavily overrepresented in the feature space coverage. While this might be a minor issue, using NDVI SD twice seems arbitrary given the wide range of features employed in this study. Was there a specific rationale behind this

choice? To a minor degree this also applies to using Flow acc. twice at different scales but at least they appear to be less correlated.

*Reply:* Yes, the NDVI SD dataset are correlated to a certain degree, but show different pattern related to local variability, which is what we are aiming at with the sampling design.

(3) Including a correlation matrix of the selected features in an Appendix could be useful. This addition may also help address some of the other points mentioned.

*Reply:* Please see our reply above.

**Section 2.5.1:** Was a method used to reduce the dimensionality of the features, such as a correlation matrix filter, feature elimination, PCA, or a similar technique?

*Reply:* No. Please see our reply to your comment on L. 233 below.

**L. 227:** A five-times repeated 10-fold cross-validation (CV) has been applied. However, the methodology seems to suggest that a non-nested CV was used, despite the fact that hyperparameters were tuned. This approach is likely to result in slightly overoptimistic results. Although the caret package does not offer nested CV by default, using a single nested 10-fold (outer) and 5-fold (inner) CV would require the same computational resources given the five-times repeated 10-fold CV. Implementing a nested CV would ensure the independence of the test data during hyperparameter tuning, which is particularly important as the predictions will be used for the pedotransfer function.

*Reply:* We have just tested that. The differences are marginal and based on different sample set sizes and hence not 100% comparable. Because single 10-fold CV is usually not stable, it should at least be a 5-times nested 5-times 10-fold CV. This is computationally very demanding especially when using more than one ML algorithm. Hence, the neglectable difference in accuracy as well as the computational burden must be the reason nested CV is rarely used (see Piikki et al. 2021).

**L. 233:** As a recommendation, in case the modeling is repeated, consider that 600 features represent a large number relative to the sample size. Without feature selection, the sample-to-feature ratio is nearly 1:1, which heightens the risk of overfitting.

*Reply:* Usually, accuracy increases a bit when smaller feature set sizes are used. What we often see with more covariates is reduced visual artifacts in the resulting maps. Depending on the ML model, the number of features can have a strong influence on computation time. Therefore, we use decorrelation in projects where we use many more features (> 1000), even though high correlation does not always mean redundancy.

**L. 234:** It is unclear how the "pedotransfer function" is implemented. Are the predictions of the other soil properties used as additional features for the second model? If so, are they used according to the same "training fold" splits? Additional code and/or more detailed information would be necessary. Particularly, if there's leakage during

hyperparameter tuning or if different "training folds" are used, the results may be too optimistic.

*Reply:* Yes, we use them as additional predictors. We also transparently discuss the improvement: "Due to the inter-correlation between the soil fractions, the respective results must be treated cautiously." (L303-304). Data leakage in the CV, if this generally plays any role, should not be relevant.

**Section 2.5.3:** Were the models evaluated based on the same test folds? Otherwise, comparability is slightly limited and subject to randomness from the splitting.

Models are evaluated based on a non-probability sampling design. Given the large number of samples and the spread in geographic space, this may not be a large issue but could be a point to consider (see Piikki et al. 2021).

*Reply:* Based on repeated CV this should not matter. Thanks for the reference! See also Wadoux et al. (https://doi.org/10.1016/j.ecolmodel.2021.109692).

**Piikki, K., Wetterlind, J., Söderström, M., & Stenberg, B. (2021).** Perspectives on validation in digital soil mapping of continuous attributes—A review. Soil Use and Management, 37(1), 7-21.

**Results**

The study proposes an interesting framework in which various methods are combined. The results are promising given the high $R^2$. However, this alone is still not convincing that the framework is actually "capable". It would be useful to have a reference performance. E.g., what if only ordinary kriging is used compared to all the different feature engineering during the spatial modelling or what if samples are selected randomly instead of this "complicated" sub-sampling approach based on locality? Evidence that could demonstrate an increase in accuracy with the proposed framework would benefit the paper significantly.

*Reply:* Do we really have to compare ML based approaches to ordinary kriging nowadays? An ML approach relies on structural dependencies, i.e. on soil forming factors (clorpt or scorpan), representing the cause of soil formation. Kriging is based on spatial dependencies, which are symptoms but never the cause. Also, neglecting any covariate information will result in wrong models. This was shown in various publications.

We do not want to show that an ML approach works. We used this approach to do the best we can to generate good maps. Usually, most publications focus on one method (e.g. random forest). This can be for a purpose if the method has specific features (e.g. quantile regression forests). But often publications present some new or adapted method somewhere in the chain. Here, we focus on getting it operational. Therefore, we

want a stable system and combine promising methods. An in-depth analysis of all possible combinations is simply not feasible. We show that combining the methods is promising.

**Section 3.2:** The sampling design is an integral part of this paper and the reader is supposed to believe that this new sampling design is more efficient than other common sampling designs. However, this results section does not contextualize the new sampling design compared to other commonly used sampling designs. Without evidence and comparisons to another sampling designs, it is simply not convincing for the reader, that this new design is actually capable of improving predictions.

*Reply:* As mentioned, it is not exactly about improving predictions. The aim was a sampling design, that covers the local fine scale variability and to be able to systematically subsample from that design to find a sample set size required. Once we know how many samples are required (after multiple further projects) we can then switch to another design or compare this one on a lower density to others.

**L. 258 – 259:** "*A comparison of the frequency distributions of the input data set (grids) with that of the selected sampling locations reveals a high degree of correspondence (Figure 10)*"

Is this supposed to be a "good" thing?

*Reply:* We think that this is a good thing and it's also an aim of many sampling design approaches. Otherwise, they would not really cover the feature space.

Random Sampling does this probability the best. In contrast, with feature coverage, one may even expect deviation from the actual frequency distribution function.

*Reply:* This depends on the sampling design algorithm and is usually not intended. KS is a bit different.

**Conclusion**

**L. 330 – 336:** This part feels out of place in the conclusion. The context of using soil property maps prior to pedological fieldwork have not been subject of this manuscript apart from the first paragraph of the introduction. In order to keep the conclusion more in touch with the actual discussion and results section, the conclusion should not reiterate the first paragraph of the introduction. L. 337 seems to be a much more appropriate start of the conclusion.

*Reply:* We will remove this paragraph.

**L. 338 – 339**: "Developing an effective sampling design is one of the most critical aspects of operationalizing soil mapping."

The biggest drawback of this study is that the "effectiveness" of the proposed sampling design has not been demonstrated but it just proposes a new framework of a sampling

design within a framework for creating soil property maps. Ultimately, there is no evidence that the new sampling design is "effective".

*Reply:* Yes, "effective" is not the right term, we will clarify this.

**Additional comments**

**L. 24**: "Study area" instead of "data set"?

*Reply:* With data set, we mean the covariates.

**L. 113**: White space is missing between "*[…] content(Safanelli […]*".

*Reply:* Thanks!

**L. 220**: Modeling should be uppercase.

*Reply:* Thanks!

**Fig. 18 – 22** may be arranged to a single Figure as it allows better evaluation and comparison.

*Reply:* We will move the data in tables.

Some Figures could be added to the Appendix because they may be interesting but do not contribute to the results or discussion (e.g. Fig. 12, 17 & 18)

*Reply:* Done.